# Effect of Gelatin Coating and GO Incorporation on the Properties and Degradability of Electrospun PCL Scaffolds for Bone Tissue Regeneration

**DOI:** 10.3390/polym16010129

**Published:** 2023-12-30

**Authors:** Carlos Loyo, Alexander Cordoba, Humberto Palza, Daniel Canales, Francisco Melo, Juan F. Vivanco, Raúl Vallejos Baier, Carola Millán, Teresa Corrales, Paula A. Zapata

**Affiliations:** 1Departamento de Ciencias del Ambiente, Facultad de Química y Biología, Universidad de Santiago de Chile (USACH), Grupo Polímeros, Santiago 9160000, Chile; cloyo@yachaytech.edu.ec (C.L.); alexander.cordoba@usach.cl (A.C.); 2School of Chemical Sciences and Engineering, Yachay Tech University, Hda. San José s/n y Proyecto Yachay, Urcuquí 100119, Ecuador; 3Departamento de Ingeniería Química, Biotecnología y Materiales, Facultad de Ciencias Físicas y Matemáticas, Universidad de Chile, Beauchef 851, Casilla 277, Santiago 8370459, Chile; hpalza@ing.uchile.cl; 4Departamento de Ingeniería Mecánica, Facultad de Ingeniería, Universidad de Santiago de Chile (USACH), Santiago 9160000, Chile; daniel.canales@usach.cl; 5Departamento de Física y Soft Matter Research Center (SMAT-C), Universidad de Santiago de Chile (USACH), Av. Victor Jara 3493, Santiago 9160000, Chile; francisco.melo@usach.cl; 6Facultad de Ingeniería y Ciencias, Universidad Adolfo Ibáñez, Viña del Mar 2580335, Chile; juan.vivanco@uai.cl; 7Facultad Artes Liberales, Universidad Adolfo Ibáñez, Santiago 7911328, Chile; 8Facultad Artes Liberales, Universidad Adolfo Ibáñez, Viña del Mar 2580335, Chile; carola.millan@uai.cl; 9Grupo de Fotoquímica, Departamento de Química Macromolecular Aplicada, Instituto de Ciencia y Tecnología de Polímeros, C.S.I.C., Juan de la Cierva 3, 28006 Madrid, Spain; tcorrales@ictp.csic.es

**Keywords:** electrospinning, polycaprolactone, graphene oxide, gelatin coating degradability

## Abstract

Polymer-based nanocomposites such as polycaprolactone/graphene oxide (PCL/GO) have emerged as alternatives for bone tissue engineering (BTE) applications. The objective of this research was to investigate the impact of a gelatin (Gt) coating on the degradability and different properties of PCL nanofibrous scaffolds fabricated by an electrospinning technique with 1 and 2 wt% GO. Uniform PCL/GO fibers were obtained with a beadless structure and rough surface. PCL/GO scaffolds exhibited an increase in their crystallization temperature (Tc), attributed to GO, which acted as a nucleation agent. Young’s modulus increased by 32 and 63% for the incorporation of 1 and 2 wt% GO, respectively, in comparison with neat PCL. A homogeneous Gt coating was further applied to these fibers, with incorporations as high as 24.7 wt%. The introduction of the Gt coating improved the hydrophilicity and degradability of the scaffolds. Bioactivity analysis revealed that the hydroxyapatite crystals were deposited on the Gt-coated scaffolds, which made them different from their uncoated counterparts. Our results showed the synergic effect of Gt and GO in enhancing the multifunctionality of the PCL, in particular the degradability rate, bioactivity, and cell adhesion and proliferation of hGMSC cells, making it an interesting biomaterial for BTE.

## 1. Introduction

Advancements in bone tissue engineering (BTE) have ushered in a new era of possibilities for restoring and regenerating damaged or lost bone tissues through the development of synthetic scaffolds that mimic the microenvironment of the extracellular matrix (ECM) of natural bone tissue scaffolds [1]. To facilitate the attachment, proliferation, and differentiation of osteogenic cells, these scaffolds must be designed with optimal mechanical, hydrophilicity, bioactivity, morphology, biocompatibility, and biodegradability properties [2,3]. Among the different kinds of scaffolds, fibrous structures produced by electrospinning stand out because they can resemble the morphology of the ECM, making them one of the most cost-effective and versatile techniques [4]. Electrospinning can use a wide range of polymeric materials for designing scaffolds with desirable characteristics, such as a high specific area and aspect ratio and interconnected pores [5,6]. Indeed, electrospinning allows the fabrication of scaffolds with fiber diameters spanning from nanometers to a few micrometers, thus yielding a high surface-to-volume ratio and mechanical strength closely mimicking the ECM [7].

Among the different polymeric materials used to obtain electrospun scaffolds, poly (ε-caprolactone) (PCL), a semi-crystalline polymer, has been emphasized for BTE [8] due to its advantageous mechanical properties, including its excellent resistance to deformation and high flexibility, as well as its approval by the US Food and Drug Administration [9]. Furthermore, PCL is soluble in several organic compounds, making it highly suitable for electrospinning processes in biomaterial research [10]. Furthermore, PCL is considered non-toxic, and its breakdown products resulting from degradation can be safely eliminated through normal metabolic processes [11,12]. However, PCL has low bioactivity and cellular affinity, and this has motivated the development of different high-performance PCL nanofiber scaffolds that present enhanced mechanical properties, hydrophilicity, and biological responses that can increase hard tissue repair [13].

Graphene oxide (GO), a layered material derived from graphene, holds great promise as a filler in polymer nanocomposites not only because of its high reinforcement effect but also because of its large number of functional groups, including epoxide, carboxyl, and hydroxyl, enhancing the adsorption of proteins from the ECM and stem cell adhesion, proliferation, and differentiation [14,15,16,17]. In PCL, GO drastically improves the stiffness of the polymer at low concentrations [18]. Notably, PCL/GO scaffolds prepared by electrospinning increase the cell attachment, proliferation, infiltration, and differentiation of different stem cells [13,19]. In these electrospun scaffolds, GO can further act as a reinforcement agent [13].

PCL has also been combined with natural biopolymers, such as gelatin (Gt), to improve its low hydrophilicity and biodegradation, further rendering biological information guidance to cells [3]. Gt is a protein obtained from collagen, a major component of the native ECM, presenting exceptional cell adhesion, proliferation, and differentiation properties, along with biocompatible and biodegradable characteristics [20,21]. PCL/Gt composites have been prepared by electrospinning with a blend of PCL and Gt solutions. Biological studies have shown that PCL/Gt scaffolds support cell attachment, promote cell growth, and potentially facilitate the development of musculoskeletal tissues [22]. However, the addition of this protein led to a reduction in both Young’s modulus and the maximum tensile stress compared with the values obtained with pure PCL scaffolds. The incorporation of Gt into PCL further improved wettability, leading to the improved adhesion and proliferation of MC3T3-e1 cells and the promotion of bone formation, as evidenced by in vitro osteogenesis characterization [20].

Motivated by the excellent behavior of both GO and Gt fillers, PCL ternary compounds have been studied to create electrospun scaffolds with appropriate biological properties [23,24]. For instance, electrospun PCL/Gt/GO scaffolds with 5 wt% tetracycline hydrochloride (TCH) present high hydrophilicity and a suitable microenvironment for cell migration, adhesion, and proliferation [25,26]. Unal et al. proposed an innovative method for the fabrication of PCL/Gt/GO scaffolds using an electrospun PCL scaffold electrosprayed with Gt and GO at varying concentrations (0.05, 0.1, and 1.0 wt%). The incorporation of 1 wt% GO resulted in a 110% increase in mechanical strength compared with that of the PCL/Gt scaffold. Moreover, the PCL/Gt and PCL/Gt/GO nanofiber membranes demonstrated favorable biocompatibility, biodegradability, and interaction with U251 MG cells [21]. Examining the performance of higher concentrations (1.0 wt%) of GO is crucial for developing scaffolds with enhanced thermal and mechanical properties while ensuring non-toxicity as well as evaluating the effect on degradability over time.

Besides mixing PCL with biopolymers or nanoparticles, another strategy to improve its bioactivity toward BTE is by coating it, particularly with Gt, as this provides desirable biological properties to the bulk polymer while maintaining its structural properties [27,28]. An electrospun PCL scaffold coated with Gt showed improved mechanical behavior compared with PCL/Gt composites, and its bioactivity was higher than that of pure PCL, although it was lower than that of the composites [27]. Three routes have been explored to coat PCL with Gt: direct coating, coating on a surface-modified PCL (such as by plasma treatment), and coating followed by a crosslinking Gt reaction [28,29,30]. Crosslinking and surface modification processes are currently being explored to delay the high solubility of Gt in aqueous solutions, with the former being the most commonly used, although it can increase the toxicity of the scaffolds [31] and does not completely avoid the solubility of Gt [32]. Although the different functional groups of GO can increase the polarity of PCL and, therefore, the adsorption of biomolecules and biopolymers, there are no studies yet about the effect of this filler on the Gt coating characteristics of PCL/GO scaffolds. 

As mentioned above, the existing literature has primarily reported the formation of integrated PCL/Gt/GO through the electrospinning of blended compounds. In the current study, we employed the electrospinning technique to obtain PCL/GO fibers, followed by a direct coating with Gt without a crosslinking process, allowing us to quantify the effect of GO on the absorption and solubility of Gt. We hypothesized that the presence of GO in the PLC would enhance the absorption of Gt, ensuring its direct exposure on the surface for longer periods and optimizing its efficacy without any chemical treatment.

Our research further aimed to assess the impact of the Gt coating on other scaffold properties, including water absorption, degradability, and thermal and mechanical characteristics. Additionally, we evaluated the potential use of these coated scaffolds for BTE by examining their in vitro bioactivity, viability, and biocompatibility, as determined by cellular adhesion and proliferation. Through this comprehensive analysis, we aimed to gain a deeper understanding of the potential applications of Gt-coated PCL/GO in promoting efficient bone tissue regeneration.

## 2. Materials and Methods

### 2.1. Materials

Polycaprolactone (PCL; Mw 80,000 g/mol) and bovine skin Gt Type B were purchased from Sigma Aldrich, Darmstadt, Germany. ACS-grade chloroform (CF), dimethyl sulfoxide (DMSO; max. 0.025% H_2_O), and LC–MS-grade water for chromatography were purchased from Merck, Darmstadt, Germany. Fine graphite powder, sodium nitrate (>99%), potassium permanganate (>99%), sulfuric acid (>95%), and hydrogen peroxide (30%) were purchased from Sigma-Aldrich, Darmstadt, Germany. Resazurin sodium salt (80% dye content) was purchased from Santa Cruz Biotechnology, Dallas, Texas, USA. Polydimethylsiloxane (PDMS) was purchased from Dow Corning Corporation, Midland, Michigan, USA, and high-glucose Dulbecco’s Modified Eagle Medium was purchased from Gibco, Grand Island, NY, USA.

### 2.2. Graphene Oxide Synthesis

Graphene oxide (GO) was prepared through a modified Hummer method, as previously reported by Olate-Moya et al. (2020) [33]. Briefly, 1.0 g of graphite powder and 0.5 g of sodium nitrate were mixed, and then 25 mL of concentrated sulfuric acid was added under constant magnetic stirring for 30 min. To ensure a controlled temperature (below 20 °C), an ice bath was employed while gradually adding 3.0 g of KMnO_4_ to the mixture, which was then stirred for an additional 2 h at 35 °C. 

Afterward, 500 mL of water was added, and the solution was stirred for 1 h. To terminate the reaction, 5 mL of H_2_O_2_ solution (30%) was added. The solution was then centrifuged, washed with HCL and water, and filtered. Finally, the obtained solid residue was suspended in deionized water and subjected to mechanical exfoliation for 1 h using a Vibra-Cell VC 505 ultrasonic processor (Sonics & Materials, Newtown, CT, USA). The suspension was then freeze-dried to obtain the GO films. 

### 2.3. Scaffold Fabrication by Electrospinning

To obtain PCL fibers, a CF/DMF (90/10 *v*/*v*) solvent mixture was utilized. The PCL was mixed with the binary solvent and stirred for 24 h at 360 rpm to achieve a concentration of 12.5 wt/V%. The polymer solution was then electrospun using electrospinning equipment (Tong li Tech, TL-01, Shenzhen, China) at room temperature. The polymer solution was supplied at a constant flow rate of 2 mL/h and a voltage of 20 kV. The fibers were collected on a stainless steel roller placed 16 cm away from the needle tip (20G). Subsequently, the obtained scaffolds were stored in a desiccator for 48 h to ensure the removal of any residual solvent and refrigerated at 4 °C until characterization.

To prepare the PCL/GO scaffolds, the synthesized GO was first dispersed in chloroform to achieve concentrations of 1 and 2 wt% relative to the PCL. The GO dispersion was then mixed with the PCL in the binary solvent system and stirred for 24 h. Subsequently, the dispersion was sonicated for 1 h. Finally, the PCL/GO dispersions were electrospun under the same conditions as the neat PCL. The scaffolds were labeled PCL, PCL/1% GO, and PCL/2% GO, corresponding to 0, 1, and 2% GO.

### 2.4. Gelatin Coating

To carry out the Gt coating, a 2.5% *w*/*v* solution of Type B Gt in distilled water was prepared. Subsequently, the scaffolds were immersed in this Gt solution for 48 h at 37 °C. Finally, the scaffolds were dried in an oven for 1 h and stored refrigerated at 4 °C until characterization.

### 2.5. Characterization of Electrospun Scaffolds

#### 2.5.1. Morphology

The morphology of the electrospun fibers was examined in a field emission scanning electron microscope (FESEM) operating at 5 kV (Quanta, FEG 250, Waltham, MA, USA). Prior to imaging, the samples were coated with gold using a sputtering coater to enhance conductivity. The fiber diameter distribution was determined by measuring about 100 fibers using Image-J 1.53e software, Bethesda, Maryland, USA.

#### 2.5.2. Infrared Spectroscopy Analysis

The chemical nature of the materials was analyzed using Fourier-transform infrared spectroscopy with attenuated total reflection (ATR-FTIR) on a PerkinElmer Spectrum Two instrument, Llantrisant, UK. FTIR spectra were taken in the range of 400–4000 cm^−1^ with a resolution of 4 cm^−1^, an accumulation of 16 scans, and a step increment of 1 cm^−1^.

#### 2.5.3. Raman Spectroscopy Analysis

The Raman measurements were carried on a confocal microscope Raman Witec Alpha 300, Oxon, UK. An Ar laser with a 532 nm excitation wavelength and a 20× microscope objective with a numerical aperture of 0.75 were used. The laser power on the samples was about 1.4 mW, the spectral resolution was 4 cm^−1^, and 32 scans were performed per second. The spectra were recorded over the 1000–3800 cm^−1^ region.

#### 2.5.4. Water Absorption and Biodegradability Analysis

The water absorption and biodegradability of the scaffolds were assessed by immersing them in a PBS solution. Square scaffolds with a 1.5 cm side were cut out and placed in 50 mL tubes containing 15 mL of PBS solution with a pH of 7.4. The tubes were placed in an incubator with a shaker set at 37 °C. After specific time intervals of 1, 7, 14, 28, and 60 days, the scaffold samples were removed from the tubes, washed twice with distilled water, and subsequently dried under vacuum for 48 h. The structural integrity of the scaffolds was evaluated by FESEM. Both wet and dry samples were weighed on an analytic balance. The water absorption and weight loss were calculated using Equations (1) and (2), respectively.
(1)$$Water\;absortion\left(\%\right)=\frac{W_{wet\;sample\;t}-W_{0}}{W_{0}}*100$$
(2)$$Weight\;loss\left(\%\right)=\frac{W_{0}-W_{dry\;sample\;t}}{W_{0}}*100$$
where $W_{0}$ is the initial weight of the sample before conducting the tests, $W_{wet\;sample\;t}$ corresponds to the weight of the wet sample after a specific immersion time, and $W_{dry\;sample\;t}$ refers to the weight of the dry sample after each drying stage.

#### 2.5.5. Thermal Analysis

Differential scanning calorimetry (DSC) measurements were carried out using a Mettler Toledo DSC1/500 instrument. Samples were heated from −5 to 80 °C in a nitrogen atmosphere at 10 °C/min, held at 80 °C for 10 min to stabilize, and finally cooled to −5 °C.

The crystallinity degree (χ%) of PCL in the fibers was calculated using the following equation:(3)$$\chi_{c}=\left(\frac{{\Delta H}_{f}}{{\Delta H}_{f}^{0}\times \left(\frac{1-x}{100}\right)}\right)\times 100$$
where Δ*H_f_* is the fusion enthalpy of PCL in the fibers; Δ*H_f_^0^* corresponds to the theoretical fusion enthalpy of 100% crystalline PCL, which was considered to be 139.5 Jg^−1^ [34]; and *x* represents the weight percent of GO and Gt in the scaffolds.

Thermogravimetric analyses (TGA) were carried out using a Mettler Toledo TGA/SDTA 851 setup under a nitrogen-inert atmosphere (50 mL/min). Samples were placed in alumina crucibles and heated from 25 to 700 °C at a heating rate of 10 °C/min. The Gt content of the scaffolds was determined by comparing the residual weights at 600 °C of each material and using the following equations [35]:(4)$$\sum_{i}{Rt(\%)}_{i}^{0}\cdot {Wf}_{i}^{S}={Rt(\%)}_{total}^{}$$
(5)$$\sum_{i}{Wf}_{i}^{S}=1$$
where *Rt* (%)_i_^0^ represents the residual weight percentage of the *i-th* component, *Wf_i_^S^* is the weight fraction of the *i-th* component in the scaffold, and *Rt* (%)_total_ is the residual weight percentage of the electrospun mat with all the components.

#### 2.5.6. Mechanical Properties

The mechanical properties of the electrospun scaffolds were determined using an uniaxial tensile test conducted on CellScale BioTester 5000 equipment. Rectangular sheets with dimensions of 2.3 mm × 8.8 mm were cut from the scaffolds. The thickness of each sample was measured using a digital micrometer. During the tensile test, the specimens (n = 5) were elongated at a 10 mm/min rate with a load cell of 50 kN at room temperature. From the stress–strain curves obtained, the Young’s modulus, strain at break, and ultimate tensile stress were determined.

#### 2.5.7. In Vitro Bioactivity Analysis

The bioactivity of the scaffolds was determined by observing the formation of hydroxyapatite on the surface of the fibers during their immersion in a simulated body fluid (SBF) solution for 28 days, following the procedure of Kokubo et al. (2006) [36]. The scaffolds were cut into rectangular shapes measuring 1.5 cm on each side and placed in a Falcon tube containing 15 mL of SBF solution. The containers were incubated at 37 °C for 28 days [36]. After 14, 21, and 28 days, the samples were analyzed by ATR-FTIR to examine the changes in their chemical composition. Additionally, at the end of the 28-day period, the samples were analyzed by FESEM coupled with energy-dispersive X-ray spectroscopy (EDX) to investigate their surface morphology and elemental composition.

#### 2.5.8. Scaffold Biological Assessment

Human gingival mesenchymal stem cells (hGMSCs) were used to test the biological performance of the scaffolds [37,38]; they were acquired and cultured following our previously reported methods [39,40]. hGMSCs from primary cell cultures were obtained from human participants after approval by the Institutional Review Board of Universidad Adolfo Ibáñez, IRB-UAI, and the Bioethical Research Committee (approval number 54/2019), as reviewed in Benjumeda et al. 2020 [37]. All experiments associated with cell analysis were performed under the Manual of Biosafety Standards and Associated Risks (Fondecyt-CONICYT 2018) and approved by IRB-UAI.

Electrospun scaffolds were placed in specialized holders, which were manufactured in polylactic acid in an in-house 3D printer [39]. The holders were sterilized in EtOH 70% overnight, and the scaffolds were sterilized under UV light for at least one hour per side [41]. 

hGMSCs from passages 7–9 were trypsinized and counted in a Luna-II automated cell counter (Logos Biosystems, Dongan-gu Anyang-si, Gyeonggi-do, South Korea). One positive and two negative controls were used. The positive control consisted of cells directly added to the wells of a 24-well tissue culture plate. The negative controls were culture media without cells, and both were added to each scaffold and the empty wells of the tissue culture plates (for background subtraction). For all experiments, three scaffolds were used per condition (*n* = 3), as well as for the positive and negative controls. 

To determine cellular viability and biocompatibility, we used MTS kit assays (Promega). The assays make it possible to determine viable cells by reducing tetrazolium salt (MTS) to formazan in metabolically active cells. The results are expressed in relation to the cells that adhered to the substrate (scaffold) and the number of cells that proliferated on the substrate (scaffold).

##### Cell Adhesion Efficiency

Cell adhesion efficiency was evaluated as previously reported [39,40]. Briefly, 4.5 × 10^4^ hGMSCs in 200 µL of culture media were added on top of each scaffold and placed in each holder for two hours (together with the controls). Later, 1 mL of culture media was added to each well of the 24-well culture plate and kept under standard conditions overnight. The next day, the scaffolds were transferred to a new culture plate with culture media using sterile tweezers rinsed three times with 1 mL of sterile PBS (to remove non-adherent cells) and completed with 1 mL of fresh culture media, and finally, 200 µL of the CellTiter 96^®^ Aqueous One Solution Cell Proliferation Assay (MTS, Promega, Madison, Wisconsin, USA) reagent was added to each well for two hours under culture conditions. Subsequently, 200 µL of the supernatant was measured in a plate reader (Biotek 800 TS, Merck (Darmstadt, Germany) at 490 nm. Cell adhesion efficiency is expressed in relation to the control (cells in the well), which was considered to be 100% [39,42]. 

##### Cell Proliferation

Cell proliferation was evaluated according to previously reported methods, like cell adhesion efficiency, but after 3 and 7 days of culture. Briefly, 1.5 × 10^4^ hGMSCs were used and kept under standard culture conditions for 3 and 7 days. At each time point, 1 mL of fresh culture media was mixed with 200 µL of the CellTiter 96^®^ Aqueous One Solution Cell Proliferation Assay reagent (MTS, Promega, Madison, Wisconsin, USA). Subsequently, 200 µL of the supernatant was measured in a plate reader (Biotek 800 TS, (Darmstadt, Germany) at 490 nm. The results are shown in relation to the optical density (O.D.) of each sample. Cell proliferation was measured in both the positive (cells in the wells) and negative controls (culture media without cells, as well as scaffolds without cells, for background subtraction).

### 2.6. Statistical Analysis

All quantitative data are reported as means ± standard deviations. To determine statistically significant differences between groups, an analysis of variance (two-way ANOVA) was conducted, followed by Tukey’s test. A significance level of 0.05 was employed to identify statistically significant differences. The analysis focused on the effects of two parameters—GO concentration (0%, 1%, and 2%) and gelatin coating—to evaluate degradation, mechanical, thermal, and biological properties.

## 3. Results and Discussion

### 3.1. GO Characterization

GO was morphologically characterized by SEM and TEM. Figure 1A shows the SEM image, which reveals that GO exhibited a distinctly rougher surface and had thin sheets arranged in a random and overlapped fashion. Figure 1B shows the TEM image, which provides additional insights by showcasing a single, transparent, and wrinkled sheet of GO. The morphology of our product was quite consistent with that previously reported [43,44,45,46].

The structural information of GO shown in Figure 1C was obtained from the XRD spectra. The XDR GO diffractogram showed two characteristic peaks at 2θ = 11.0° and 42.3°, corresponding to the reflections from the (002) and (101) planes [47,48]. The interlayer spacing was calculated at 0.80 nm.

### 3.2. Morphology of the Scaffolds

Initially, the PCL fibers were produced using only chloroform as a solvent, resulting in noticeable agglomerations, defects, and beads, as illustrated in Appendix A. To improve fiber homogeneity, a dual solvent approach employing chloroform and dimethylformamide (DMF) at a ratio of 9:1 was adopted. By employing this method, the PCL, PCL/1 wt% GO, and PCL/2 wt% GO samples were processed, producing intricate and bead-free interconnected structures with a surface roughness attributed to the distinct boiling points of the solvents, as observed in the SEM images shown in Figure 2. A similar observation was reported by Canales et al., who showed analogous roughness behavior in electrospun PLA fibers using dichloromethane (DCM) and DMF solvents at a 9:1 ratio [8]. The combined solvents induced phase separation, resulting in heightened surface roughness of the fiber structures. The authors concluded that this topography positively influenced cell adhesion, viability, and proliferation [8].

Figure 2 also shows the fiber diameter distribution for PCL, PCL/1 wt% GO, and PCL/2 wt% GO scaffolds derived from the SEM images. The incorporation of 1 wt% and 2 wt% GO in PCL resulted in a substantial reduction in diameter by 23.2% and 28.8%, respectively, compared with neat PCL fibers (see Figure 3). This reduction in diameter serves as an indirect yet indicative measure of the successful integration of GO and is correlated with the heightened conductivity and charge density of the polymer solution [49]. Generally, an increase in viscosity leads to larger average diameters, while an upsurge in electrical conductivity tends to produce smaller average diameters [18,50]. 

In particular, the presence of GO in a PCL increases the conductivity because of an increase in sp^2^ and sp^3^ carbon domains that facilitate electron transport [25,51,52]. Indeed, Song et al. reported an increased conductivity of PCL with the incorporation of GO and explained that the nanofiber can be well elongated, split off, and separated into thinner nanofibers, which result in a smaller average diameter of the PCL/GO composite nanofibers [13,53].

### 3.3. Raman Spectroscopy Analysis

The Raman shifts of (a) PCL and (b) PCL/2 wt% GO obtained after focusing the confocal on the sample surface are presented in Figure 4. Several peaks characteristic of PCL were observed, the more prominent at 916 cm^−1^(*ν*C–COO), and others within the spectral ranges 1003–1110 cm^−1^ (skeletal stretching), 1270–1300 cm^−1^ (*ω*CH_2_), 1405–1470 cm^−1^(*δ*CH_2_), and 2800–3200 cm^−1^ (*ν*CH) refer to the crystalline fraction [54]. However, the characteristic Raman signal of GO was not detectable in the Raman spectrum with this methodology (See Figure 4b). 

To reveal the GO Raman signal, an enhancement strategy was employed; shell-isolated nanoparticles (SHINs) allow the enhancement of the Raman signal of GO by coupling with GO groups (preparation and details will be given in enhanced Raman spectroscopy applied to a study on the water–graphene interface, to be published by A. Maine, L. Caballero, and F. Melo). SHINS were deposited onto the sample surface and left to dry. Subsequently, a confocal microscope was focused on the surface where the GO spectrum exhibited its maximum intensity (Figure 4c). Thus, the GO Raman spectrum was observed, which allowed us to confirm the presence of GO in the samples. Notably, the main lines of GO corresponding to the D and G peaks were clearly visible [55]. Furthermore, in Appendix A, analogous outcomes are observed in the spectra of PCL/1wt% GO, corroborating the incorporation of GO. 

In addition, Figure 4d is slightly out of focus, which permits the observation of both Raman spectra, those of GO and PCL.

### 3.4. Gelatin Coating Scaffolds 

The SEM images of the PCL, PCL/1 wt% GO/Gt, and PCL/2 wt% GO/Gt samples (Figure 5) show the uniform and comprehensive coverage of Gt coating across all the fiber surfaces. Notably, the Gt coating maintained the roughness of the fiber texture obtained in the electrospinning process. Even though PCL exhibits hydrophobic characteristics, effective interaction occurs with the hydrophilic Gt, facilitated by the interaction between oxygen from the carbonyl group in PCL and the amine group in Gt [52]. The adsorption of Gt was enhanced by the presence of GO, and this will be explained in more detail later.

Next, we confirmed the presence of Gt. Figure 6 shows the FTIR spectrum for the Gt-coated scaffolds and includes the spectra from GO, Gt, and the uncoated samples (PCL, PCL/1 wt% GO, and PCL/2 wt% GO) for comparison. The absorption spectra of GO displayed the intensities of the bands assigned to the degenerate deformation of the hydroxyl group (-OH) at 3600 cm^−1^ and C=O stretching at 1727 cm^−1^, and the absorption peaks corresponded to the aromatic double bond C=C at 1630 cm^−1^, the hydroxyl O-H at 1400 cm^−1^, and the alkoxy group C-O at 1045 cm^−1^ [53,56]. However, due to the low amount of GO in the PCL/GO and PCL/GO/gelatin nanocomposites that developed, it was not possible to assign their signals because they were overlapped by the characteristic signals of the PCL matrix in the same range of wavenumbers. This overlapped signal effect was previously described in 3D-printed PLA/GO scaffolds, as reported by Guo et al. The authors mentioned that the characteristic peaks of GO have a minor intensity compared with PLA and suffer an overlapping effect, making them difficult to detect on the IR spectra [57].On the other hand, pure Gt shows two characteristic signals at 1650 cm^−1^, which represent the C=O stretching of amide I from the random coil and the α-helix conformations of Gt [58], and another signal at 1540 cm^−1^_,_ corresponding to amide II (N-H stretching) [59,60]. The spectra of all scaffolds displayed distinctive PCL bands, notably at 2865 cm^−1^ and 2944 cm^−1^, attributed to the symmetric and asymmetric vibrations of the -CH_2_ groups. Additionally, the spectrum featured a prominent band at 1722 cm^−1^, indicative of the C=O stretching from the carbonyl group of the ester, along with two discernible bands at 1044 cm^−1^ and 1237 cm^−1^, corresponding to the symmetric and asymmetric vibrations of the -C-O-C- linkage [58,60]. The typical bands of the CF and DMF solvents were not present in the FTIR analysis. This is an important factor in biomedical applications since the presence of residual solvents can be harmful to cells.

Figure 6B displays an in-depth examination of the FTIR spectra in the 1700 cm^−1^ to 1450 cm^−1^ range. While the uncoated samples did not exhibit any signals, the coated scaffolds revealed two slight but noticeable enhancements in transmittance corresponding to the characteristic amide I and amide II signals from Gt [60], thus providing confirmation of the coating’s presence.

### 3.5. Thermal Stability of the Scaffolds

The thermal degradation of the different scaffolds was studied by thermogravimetric analysis, as shown in Figure 7, and Table 1 summarizes the extracted data. The neat PCL exhibited a weight loss at ∼388 °C, which was attributed to the thermal decomposition of the polymer chain [18], while Gt had a weight loss at ∼318 °C due to the breaking of peptide bonds and protein chains [61]. In the curves of the coated samples, two weight losses were observed; the first one corresponded to Gt degradation at ∼318 °C, and the second one corresponded to PCL decomposition between 389 and 407 °C. The incorporation of 2 wt% GO into the PCL increased the maximum degradation temperature of the polymer matrix to 405 °C and 407 °C for coated and uncoated scaffolds, respectively. A similar behavior was reported by Song et al. in PCL/GO composites prepared by the electrospining method [13], which was attributed to the interaction between the oxygen groups of graphene and the PCL via hydrogen bonds and Van der Waals forces [52].

The amount of Gt coating on the PCL scaffolds was estimated from the TGA data, as shown in Table 1. This estimation was based on the consideration that the total degradation residue of the scaffolds at 600 °C corresponded to Gt residues since PCL and its composites did not exhibit any remaining mass at that temperature [20]. The incorporation of GO into the composite elevated the Gt content of the scaffold, which rose from 0.9% for neat PCL to 14.3% and 21.6% for 1 and 2 wt% GO, respectively. This enhancement can be attributed to the oxygenated groups in the GO, which facilitated the interaction with Gt.

### 3.6. Analysis of Water Absorption

Figure 8A,B display the results obtained for the water absorption of PCL, PCL/1 wt% GO, and PCL/2 wt% GO scaffolds and the following Gt-coated scaffolds: PCL/Gt, PCL/1 wt% GO/Gt, and PCL/2 wt% GO/Gt immersed for 24 h and over 60 days in PBS solution. The hydrophobic or hydrophilic nature of a surface is crucial for cellular performance, particularly adhesion and proliferation. A more hydrophilic surface typically enhances cell–material interactions, thereby influencing the overall functionality of the cells [62]. Hydrophilicity refers to the amount of water that the scaffold can absorb over a period of time. PCL has a hydrophobic nature due to the apolar groups in its structure and barely absorbs water, as shown in Figure 8 [63]. Remarkably, even after being immersed in a PBS solution for 60 days, the PCL scaffold showed no signs of water absorption. By incorporating GO into uncoated scaffolds, their water absorption improved, but this increase was not significant after 24 h because GO was found inside the scaffolds, exerting a low effect on the absorption of water on the surface. However, over longer periods, the presence of GO increased the water absorption of the scaffold, reaching percentages of 95 ± 34% and 151 ± 31% for PCL/1% GO and PCL/2% GO, respectively. This increase was due to some hydrolytic degradation of the PCL, which allowed the GO functional groups that contain oxygen-functional groups, such as –OH and –COOH, to be exposed on the surface of the scaffolds [56,57].

The Gt-coated scaffolds presented a much higher increase in hydrophilicity compared with the uncoated samples. For instance, in pure PCL at 24 h of immersion, the water absorption increased to 190 ± 37%. This behavior is due to the presence of hydrophilic groups from the Gt structure, the amino and carboxyl groups, which interact with the water molecules, allowing the retention of a greater amount of water than that in the uncoated scaffolds [64]. 

The water absorption of the coated scaffolds also exhibited a substantial increase with the presence of GO, particularly at 2 wt%, where it reached approximately 365 ± 23%. This escalation can be attributed to the higher quantity of Gt present in the composite scaffold. The incorporation of hydrophilic groups from Gt and GO in the fibers facilitated greater water penetration, leading to the enhancement of water absorption. As previously reported, the introduction of Gt and GO improved the hydrophilicity of the PCL scaffolds, which consequently resulted in improved cell–scaffold interactions [21]. 

As time elapsed, the Gt-coated scaffolds presented a decrease in the water absorption percentage, and this behavior was due to the release of Gt and scaffold degradation [65].

### 3.7. Degradability of PCL and PCL/GO with and without Gelatin Coating

Figure 9 displays the degradability assays for PCL, PCL/1 wt% GO, and PCL/2 wt% GO scaffolds and Gt-coated scaffolds immersed for 60 days in PBS solution.

The uncoated samples showed low degradability due to the semicrystalline structure and hydrophobic characteristics of the PCL [66]. The incorporation of GO into the PCL did not improve their degradability, and after 60 days, the weight loss values reached 4.7 ± 2.8%, 4.6 ± 0.9%, and 5.2 ± 2.3% for the PCL, PCL/1% GO, and PCL 2% GO scaffolds, respectively. 

The degradation mechanism of PCL has been previously described, with two predominant pathways: bulk and surface degradation. In the case of surface degradation, the diffusion process occurs at a slower rate, with water entering the polymer more gradually compared with hydrolysis, resulting in a gradual thinning of the material [67].

The degradation rate of the Gt-coated scaffolds was much higher because the surface Gt was continuously released into the PBS solution by dissolution. Zhang et al. [65] determined that Gt can fully leach from PCL/Gt hybrid scaffolds in 10–20 days in water [65]. In this sense, the Gt was dissolved entirely in water, and then the mass loss was followed by a slow and relatively constant rate of degradation caused by the slow hydrolysis of the PCL molecules [24,68]. This observation was confirmed by the SEM images, which showed that the gelatin coating was no longer clearly visible (Figure 9B). However, the images revealed swollen fibers, erosion, the formation of holes, and fiber cracking. These changes were more pronounced for the PCL/2% GO/Gt scaffolds due to their higher amounts of Gt and GO. In summary, in the PCL and PCL/GO systems, the degradation was very low, as shown in Figure 9A. For these systems, the degradation process probably occurred on the surface, and due to the high hydrophobicity of the PCL matrix, the degradation was clearly delayed. On the other hand, there was an increase in the size of the surface pores of the fibers in PCL/Gt and PCL/GO/Gt (Figure 9B), which was greater for the PCL/GO/Gt system. This increase indicates the development of a surface erosion process associated with the presence and release of gelatin, which has greater hydrophilicity and allows greater interaction between the fibers and water molecules, thus accelerating the intramolecular hydrolysis processes of the PCL matrix and breaking the fibers. 

### 3.8. Thermal Properties of the Scaffolds

The DSC analyses (first cycle) of the different samples are displayed in Figure 10A, and the trends of the crystallization temperatures (T_c_) and crystallinity degrees (X_c_) are illustrated in Figure 10B. The DSC results showed that the melting temperature of PCL did not change with GO incorporation [69] but that T_c_ increased with GO (Figure 10B), which can be attributed to the nanosheets of GO acting as nucleating agents, promoting crystallization by a heterogeneous process [52,70,71] through the strong interaction between the molecular chains of PCL and GO [52]. 

Despite the increase in Tc, a reduction in Xc was observed for compounds containing GO (Figure 10B). This phenomenon could be attributed to the formation of an intercalated structure between the PCL and GO. Upon intercalation, the mobility of the PCL chain becomes constrained by the GO layers, making it challenging for the PCL to crystallize between the GO layers [72,73,74], which results in a decrease in crystallinity.

The Gt-coated scaffolds (PCL/Gt, PCL/1 wt% GO/Gt, and PCL/2 wt% GO/Gt) presented a null effect on the melting temperature and a decrease in the X_C_ in comparison with neat PCL. The latter can be attributed to the Gt-coating process carried out at 37 °C. At this temperature, PCL begins to approach its melting point, at which its crystalline structure starts to break down, ultimately resulting in a loss of crystallinity in the scaffolds [67,72]. 

### 3.9. Mechanical Properties of the Scaffolds

The tensile stress–strain curves of the scaffolds are shown in Figure 11. The incorporation of GO into PCL increased the Young’s modulus from 7.85 MPa to 10.34 and 12.79 MPa for 1 and 2 wt% GO, respectively. This behavior was due to the intrinsic rigidity of GO, which acts as an excellent filler for polymer reinforcement [70] due to the adequate dispersion of the nanosheets in the polymer matrix and the strong interaction between the nanosheets and polymer chains. This implies that a greater strain must be applied for the chain rearrangement process to occur as a response to the applied tension [52,53]. Guo et al. reported an increase in the Young’s modulus of ca. 35.6% for PLA/GO (0.1 wt%) compared with the neat PLA obtained by 3D printing. The authors mentioned that the mechanical properties of the polymer nanocomposite were related to the content and distribution of the filler [57]. The decrease in fiber diameter is another important parameter explaining the stiffer behavior of the composited electrospun mats. Kim et al. reported the preparation of electrospun fibers based on PCL with different diameter sizes and found that a decrease in fiber diameter increased fiber rigidity due to the higher crystallinity and orientation when the fiber diameter was lower [75]. Considering this, the stiffer behavior of the PCL/GO electrospun mats could also be related to a decrease in fiber diameter.

For the Gt-coated scaffolds, the Young’s modulus did not change in comparison with the neat PCL, but the value decreased in comparison with PCL/GO. The reinforcement effect of GO was not appreciable, and these reductions were mainly attributed to a decrease in the crystallinity due to the gelatin coating (Figure 10B) [76]. The gelatin coating process was executed at 37 °C, and the onset of the melting process occurred slightly before this temperature. Although the samples did not undergo complete melting at this threshold, they experienced a reduction in crystallinity. This phenomenon was likely due to the higher mobility of polymeric chains, probably enabling the accumulation of GO nanolayers within the fibers during treatment. The resultant decline in crystallinity, coupled with the accumulation of GO, contributed to a reduction in the mechanical properties, encompassing Young’s modulus, strain at break, and tensile strength. This effect mirrors a similar annealing phenomenon observed in prior studies [76].

Different results have been reported for electrospun PCL/GO/Gt. The incorporation of GO (at 1, 1.5, and 2 wt%) increased the tensile strength and Young’s modulus by ca. 117% and 128%, respectively, when 1.5% graphene was incorporated into the PCL/Gt nanofibers compared with the values of PCL/Gt nanofibers. The authors indicated that gelatin could act as a compound for graphene functionalization or bridge the attachment of PCL and graphene [26]. The tensile strength did not change with the GO incorporations, as expected for PCL composites with these amounts of GO, as previously reported [21]. However, PCL/Gt and PCL/GO/Gt showed decreased tensile strength in comparison with the neat PCL. Unal et al. prepared PCL and PCL-Gt and showed that Gt incorporation reduced the mechanical strength of the composite, an effect attributed to the weak physical properties of gelatin [21].

However, the PCL, PCL/1 wt% GO, and PCL/2 wt% GO scaffolds increased the strain at break with the GO content in the uncoated scaffolds. Castilla-Cortazár et al. observed that GO nanosheets tended to stack within the polymeric matrix, and when effort was exerted, these were unstacked, allowing a greater deformation before the material suffered a break [70]. For Gt-coated scaffolds, the strain at break did not change in comparison with neat PCL.

### 3.10. In Vitro Bioactivity of the Scaffolds 

Figure 12 shows the bioactivity results obtained after 28 days of immersion in SBF for the different samples. For PCL, PCL/1 wt% GO, and PCL/2 wt% GO scaffolds, some small agglomerations of compounds were observed on the surface of the fibers. These agglomerations exhibited a random geometric shape, deviating from the spherical structure typically associated with hydroxyapatite. EDS confirmed that these accumulations were composed of calcium, phosphorus, magnesium, and potassium, elements that were present in the SBF solution and precipitated onto the fibers. Lui et al. [15] demonstrated that graphene oxide alone was not capable of inducing hydroxyapatite mineralization after the immersion of GO in SBF for 14 days [15]. The results, verified by SEM, FTIR, and EDX analysis, showed only a trace amount of calcium.

Conversely, the Gt-coated scaffolds (PCL/Gt, PCL/1 wt% GO/Gt, and PCL/2 wt% GO/Gt) presented distinctive spherical structures on the polymer surface, closely resembling hydroxyapatite crystals. Similar results were reported by Li et al., showing hydroxyapatite formation on PCL fibers coated with Gt after immersion in SBF [3]. The increased hydrophilicity and the introduction of functional groups such as -COOH and -NH_2_ from Gt could explain the effective calcium phosphate deposition, resulting in local supersaturation and nucleation of crystallites [3,77]. In the case of PCL/GO/Gt, the surface exhibited a more pronounced growth of crystals. A similar behavior was reported by Wan et al., who studied the effect of GO incorporation into Gt using the solution casting method [78]. 

The spectra obtained by energy-dispersive X-ray spectroscopy (EDX) coupled to SEM (Figure 12) showed that the spherical structures present on the scaffold fibers effectively correspond to hydroxyapatite, with a Ca/P ratio between 1.56 and 1.69, which is close to the Ca/P ratio of biological hydroxyapatite (1.67) [77]. Human biological apatite is always calcium-deficient, and hydroxyapatite NPs contain low percentages of carbonate, acid phosphate, sodium, and magnesium ions due to continuous contact with a flow of trace ions [15]. We also observed other elements, such as Mg, Na, and Cl, which are mineral ions present in the simulated body fluid solution [3].

Hydroxyapatite growth was also verified by ATR-FTIR analysis, as depicted in Appendix A, showing that the Gt-coated scaffold displayed a peak at 1090 cm^−1^, which is attributed to calcium phosphates, thus confirming the formation of hydroxyapatite crystals on the surface of the fibers. Phosphate groups are the precursors for the formation of hydroxyapatite, which possesses a more complex crystalline structure [78]. In the spectra (Appendix A), it is evident that this peak intensifies over time, indicating the ongoing growth of hydroxyapatite. 

Finally, alongside SEM, EDX, and FT-IR analysis, the growth of HA crystals was corroborated by X-ray diffraction (see Appendix A). Notably, for PCL/2%GO/Gt nanocomposites, after 28 days of immersion in SBF solution, a distinct peak at 32° was observed, corresponding to the (211) plane of the hexagonal HA nanocrystalline phase [79].

### 3.11. Cell Adhesion and Proliferation

In vitro biological characterization is an important analysis that should be performed in order to project a scaffold for future biomedical applications. Therefore, to determine how gelatin coating and the presence of GO nanoparticles affected the biological behavior of a PCL matrix, the effects of PCL/Gt, PCL/GO, and PCL/2 wt% GO/Gt on cell viability were studied in terms of the cell adhesion and proliferation of hGMSC and were compared with those of neat PCL. As in previous studies, the cells loaded directly onto the plate were used as controls [39,40]. Due to the similarity of the results in forming hydroxyapatite spheres in the bioactivity test, PCL/2 wt% GO was selected for the biological studies.

The cellular controls presented favorable adhesion and proliferation results because the growing conditions on the culture plates were highly optimized. Cell adhesion tests after 24 h are shown in Figure 13A and indicate that the neat PCL scaffold promoted cell adhesion, reaching values of up to 40%. Previous studies on PCL-based matrices for biomedical applications have reported that they serve as a substrate that effectively promotes both cell adhesion and proliferation [80]. Moreover, electrospun PCL improves this behavior due to the porous and interconnected structure that favors the cell adhesion and proliferation processes attributed to the nanometric structure, which increases the surface area and replicates the extracellular matrix (ECM) [81]. With the incorporation of GO, viability and biocompatibility effects were observed. The adhesion efficiency was within the range of 20-40% with respect to the controls. There was a slight increase in the average cell adhesion in the GO scaffolds without gelatin with respect to the PCL. This increase may be associated with the fact that GO increases hydrophilicity because it contains oxygenated hydrophilic groups, such as hydroxyl and carboxyl groups, which favor the absorption of proteins, including serum proteins that can affect cell adhesion [13]. On the other hand, it is important to highlight that the reduction in fiber size for PCL/GO is also a parameter that influences the improvement of cellular activity. As reported in electrospun fibers, a reduction in fiber size generates a topographic signal that the cells can recognize, and therefore, they benefit from this process [82]. Therefore, under all the conditions studied, the scaffolds allowed the viability of hGMSCs, which adhered to the substrate surfaces at different efficiency values. 

The gelatin coating did not affect the initial adhesion of hGMSCs to the PCL scaffolds. There were no significant differences in cell adhesion to the PCL/GO/Gt scaffolds compared with PCL in the samples without gelatin. However, when gelatin coating was incorporated, cell adhesion to PCL/GO decreased.

In the adhesion process, the presence of gelatin with GO significantly decreased the colorimetric signal (Figure 13A, right). As previously reported, there are differences in the proliferation of MSCs depending on the concentration of gelatin used in the culture. For instance, in a similar study, Khan et al. used adipose-derived mesenchymal stem cells (Ad-MSCs) and cultured them in vitro with different concentrations of gelatin for 24 h [83], and they found that 0.5% gelatin significantly increased the proliferation rate, as evaluated by a colorimetric assay. Conversely, 2% and 4% gelatin decreased cell proliferation significantly. In the current study, we used 2.5% gelatin and observed the same trend at 24 h, namely, a decreased colorimetric signal, which indicated a decrease in both adhesion and proliferation. Furthermore, the current research was a preliminary study that evaluated the biocompatibility effect of Gt on the PCL/GO scaffolds, and the current results will help us to establish further experimental designs. 

In addition, high concentrations of GO can affect cell viability [73]. Since gelatin is viscous and has low solubility [84], we speculate that reactive oxygen species (ROS) could be trapped within the gelatin matrix, negatively affecting cell viability in PCL/GO/Gt scaffolds. Future live/dead staining experiments will be considered to test this hypothesis and optimize GO and gelatin concentrations to evaluate their biological effects.

In turn, the effects of Gt and GO disappeared during the first 24 h (adhesion). Therefore, the cell adhesion of MSCs to a scaffold with GO and Gt would involve fixation, extension, or propagation mechanisms, the organization of the actin cytoskeleton, and the formation of focal adhesions. To the best of our knowledge, there are no studies that have specifically analyzed all these processes according to molecular interactions, and this interesting approach could be considered for future experiments to precisely determine the interaction mechanisms of MSCs under these conditions.

The results of cell proliferation at 3 and 7 days (Figure 13B) showed an increase in the average cell proliferation, except for PCL/2 wt% GO with gelatin, in which case the proliferation was maintained. In general, the scaffolds generated the necessary conditions to promote both cell adhesion and proliferation. The maintenance or increase in the proliferation of hGMSCs on the scaffolds for 7 days could be attributed to the advantageous characteristics of the electrospun matrices, such as ample porosity and interconnectivity of pores, which facilitate the transport of oxygen and nutrients that are essential for the normal and proper growth of cells within the scaffold [85,86]. Conversely, in the cell proliferation assays, we observed that GO incorporation diminished slightly over time and that the number of cells decreased compared with neat PCL. These results suggest that after one week, the incorporation of GO reduces cellular capacity. In vitro tests by Faraji et al. evaluated the use of PCL/quercetin-based scaffolds with different amounts of GO in NIH3T3 fibroblasts and showed that at concentrations greater than 1.5% GO, the scaffolds decreased cell viability [87]. Other works have reported that GO nanoparticles induce cellular toxicity at high loads in different types of cell lines due to the effects of the functional groups with oxygen that, over longer times, can generate ROS, which play an unfavorable role by damaging the cell membrane and causing the release of vital components of the cell [88,89]. Therefore, from a cellular point of view, it would be necessary to evaluate lower percentages of GO and longer culture times to assess proliferation.

Finally, the impact of the Gt coating became substantially more beneficial over time and showed a significant improvement after 7 days for PCL/Gt, possibly because the cells were already fully adhered and proliferating. The PCL/GO/Gt combination, until the days analyzed, did not affect proliferation significantly. It would be interesting to observe its effect over longer time intervals during which cells could differentiate into osteogenic lineages [90]. These preliminary findings in the development of bioactive scaffolds based on PCL/GO/Gt demonstrate their potential as a viable alternative, especially with regard to the optimization of GO loading.

These preliminary cellular experiments allowed us to evaluate two key concepts of the materials: whether the cells are biocompatible and viable on the scaffold. In all cases, the scaffolds studied allowed the viability of hGMSCs with a certain degree of variability, and the substrates analyzed were biocompatible. On the other hand, GO improved the mechanical properties of the cells and their bioactivity without altering cell proliferation.

## 4. Conclusions

Electrospun fiber scaffolds based on PCL/GO were prepared and coated with Gt for bone regeneration. Our results showed that the presence of GO not only decreased fiber diameter and increased scaffold stiffness but also improved the gelatin coating process. The Gt adsorption increased from 0.9 wt% for pure PCL to 21.6 wt% for PCL/GO with 2 wt% without the need for any surface treatment of the fibers. Our approach to improving the Gt coating by mixing PCL with GO was an interesting route to promote the bioactivity capacity of PCL, as the coated composites showed hydroxyapatite crystals after soaking in SBF solution. In addition, the in vitro biological analysis using hGSMCs demonstrated that uncoated and coated PCL and PCL/GO samples were viable substrates for cell adhesion and that Gt coating improved cell proliferation. These results introduce a new route to produce Gt-coated scaffolds taking advantage of the properties of GO as a filler.

## Figures and Tables

**Figure 1 polymers-16-00129-f001:**
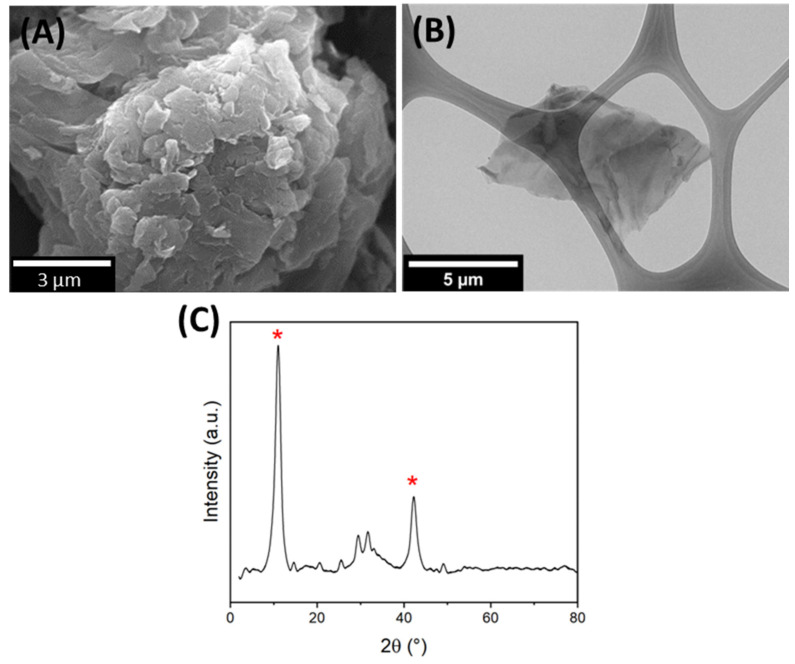
Characterization of GO by (**A**) SEM, (**B**) TEM, and (**C**) XRD, * corresponding to reflections from the (002) and (101) planes respectively.

**Figure 2 polymers-16-00129-f002:**
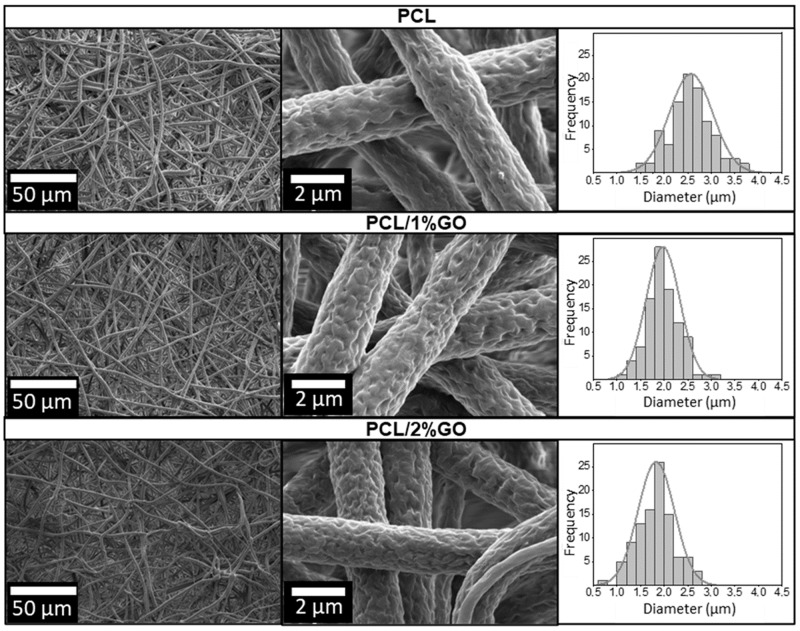
SEM image of the neat PCL fibers (**top**)**,** PCL/1 wt% GO (**middle**), and PCL/2 wt% GO (**bottom**) using DCM: DMF as solvents.

**Figure 3 polymers-16-00129-f003:**
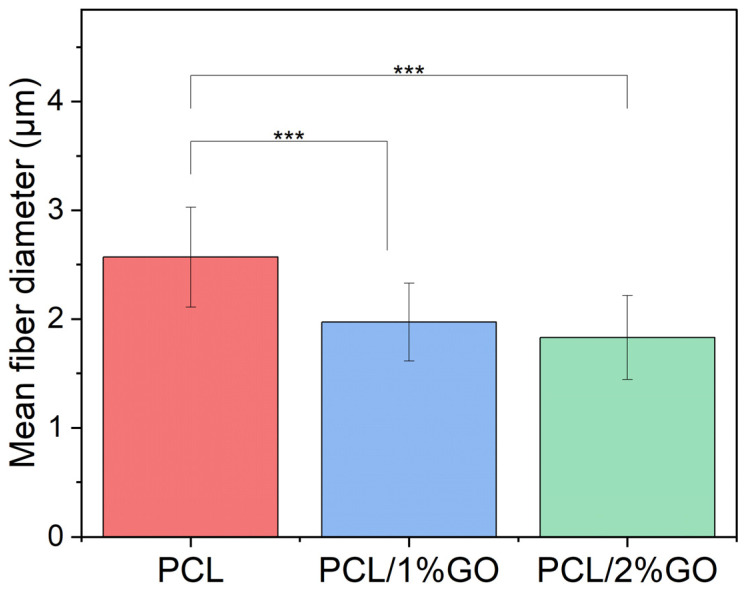
Diameters of the fibers of PCL, PCL/1 wt% GO, and PCL/2 wt% GO scaffolds. *** *p* ≤ 0.001.

**Figure 4 polymers-16-00129-f004:**
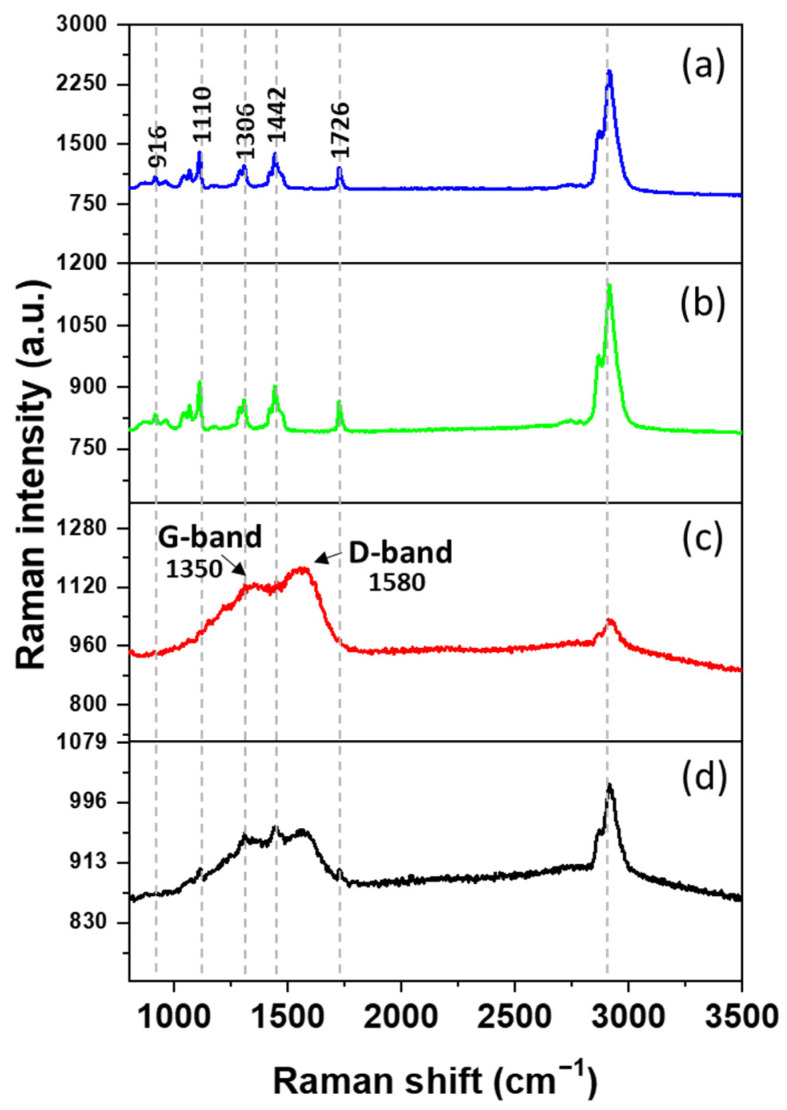
Raman intensity vs. Raman shift for (**a**) the free surface of PCL, (**b**) the free surface of PCL/2 wt% GO, and (**c**) the free surface of PCL with SHINS. (**d**) Raman spectrum taken with focal plane slightly below the free PCL/2 wt% GO surface revealing distinct spectra corresponding to both GO and PCL signals.

**Figure 5 polymers-16-00129-f005:**
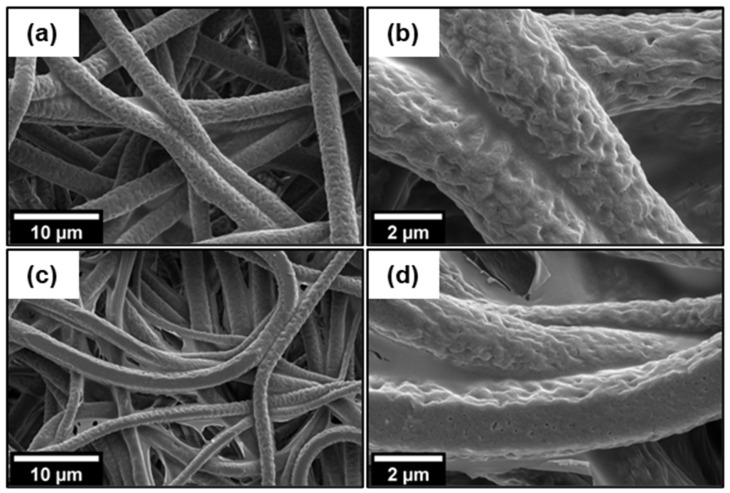
SEM images of the fibers of (**a**,**b**) PCL/1 wt% GO/Gt, and (**c**,**d**) PCL/2 wt% GO/Gt scaffolds coated with Gt.

**Figure 6 polymers-16-00129-f006:**
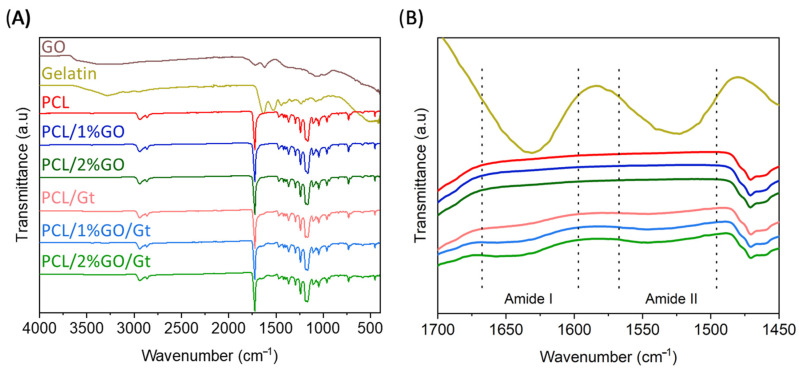
(**A**) ATR-FTIR spectra of the PCL, PCL/1 wt% GO, and PCL/2 wt% GO scaffolds and the Gt-coated scaffolds (PCL/Gt, PCL/1 wt% GO/Gt, and PCL/2 wt% GO/Gt). (**B**) Close-up view in the range of 1450 cm^−1^ and 1700 cm^−1^.

**Figure 7 polymers-16-00129-f007:**
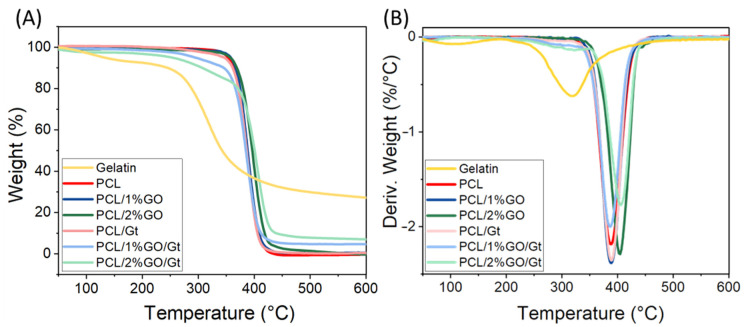
(**A**) TGA analysis under inert atmosphere. (**B**) DTA curves for PCL, PCL/1 wt% GO and PCL/2 wt% GO scaffolds and the Gt-coated scaffolds (PCL/Gt, PCL/1 wt% GO/Gt, and PCL/2 wt% GO/Gt).

**Figure 8 polymers-16-00129-f008:**
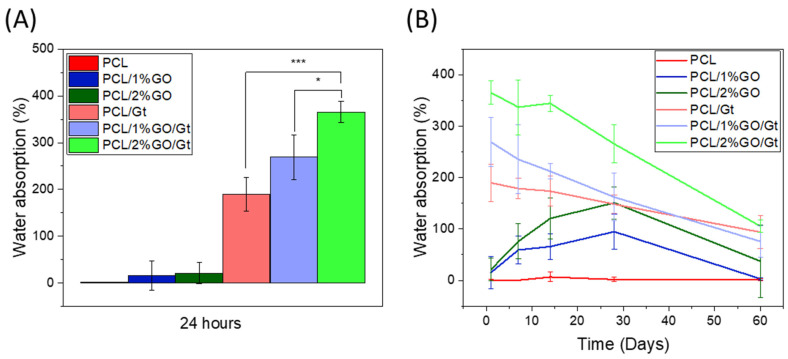
Percentage of water absorption for PCL, PCL/1 wt% GO, PCL/2 wt% GO, PCL/Gt, PCL/1 wt% GO/Gt, and PCL/2 wt% GO/Gt scaffolds immersed over (**A**) 24 h and (**B**) 60 days in PBS solution. * *p* ≤ 0.05 and *** *p* ≤ 0.001.

**Figure 9 polymers-16-00129-f009:**
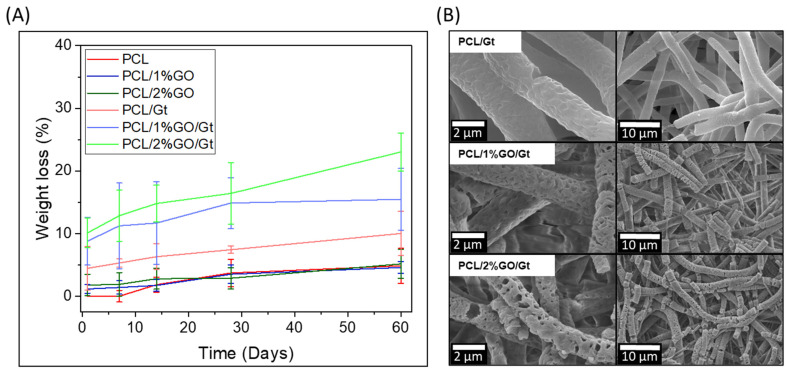
(**A**) Weight loss in the biodegradability tests for PCL, PCL/1 wt% GO, and PCL/2 wt% GO scaffolds and the Gt-coated scaffolds PCL/Gt, PCL/1 wt% GO/Gt, and PCL/2 wt% GO/Gt immersed for 60 days in PBS solution and (**B**) SEM images after 60 days of degradation.

**Figure 10 polymers-16-00129-f010:**
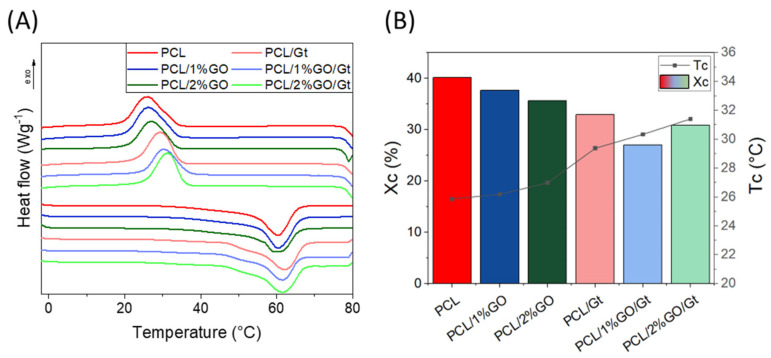
(**A**) DSC analysis (first cycle), and (**B**) crystallization temperature (Tc) and crystallinity degree (Xc) of the PCL, PCL/1 wt% GO, and PCL/2 wt% GO scaffolds and the following Gt-coated scaffolds: PCL/Gt, PCL/1 wt% GO/Gt, and PCL/2 wt% GO/Gt.

**Figure 11 polymers-16-00129-f011:**
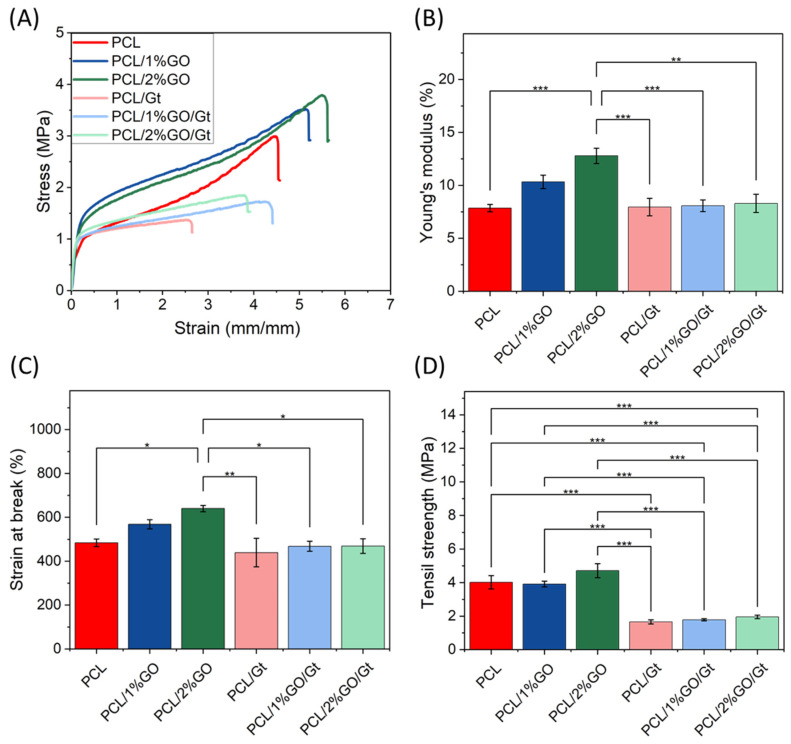
Mechanical properties of PCL, PCL/1 wt% GO, and PCL/2 wt% GO scaffolds and the following Gt-coated scaffolds: PCL/Gt, PCL/1 wt% GO/Gt, and PCL/2 wt% GO/Gt. (**A**) Tensile–strength curves, (**B**) Young’s modulus, (**C**) tensile strength, and (**D**) elongation at break. * *p* ≤ 0.05, ** *p* ≤ 0.01, and *** *p* ≤ 0.001.

**Figure 12 polymers-16-00129-f012:**
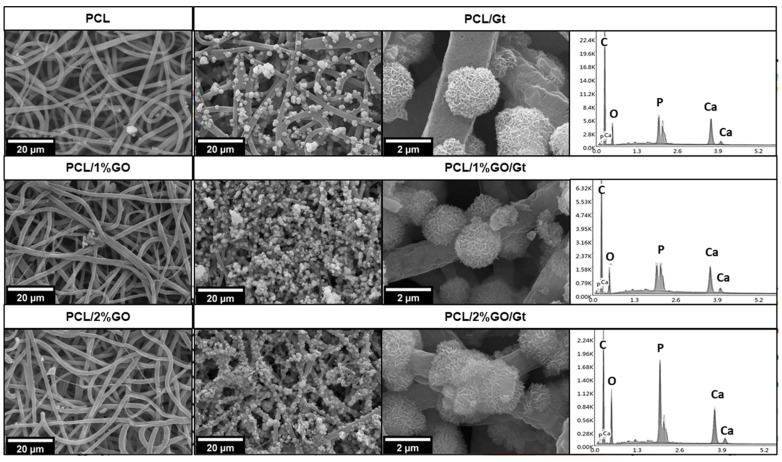
SEM images (left) of PCL, PCL/1 wt% GO, and PCL/2 wt% GO scaffolds and the Gt-coated scaffolds PCL/Gt, PCL/1 wt% GO/Gt, and PCL/2 wt% GO/Gt after 28 days of immersion in SBF solution. EDS spectra (right) of hydroxyapatite on PCL/Gt, PCL/1 wt% GO/Gt, and PCL/2 wt% GO/Gt.

**Figure 13 polymers-16-00129-f013:**
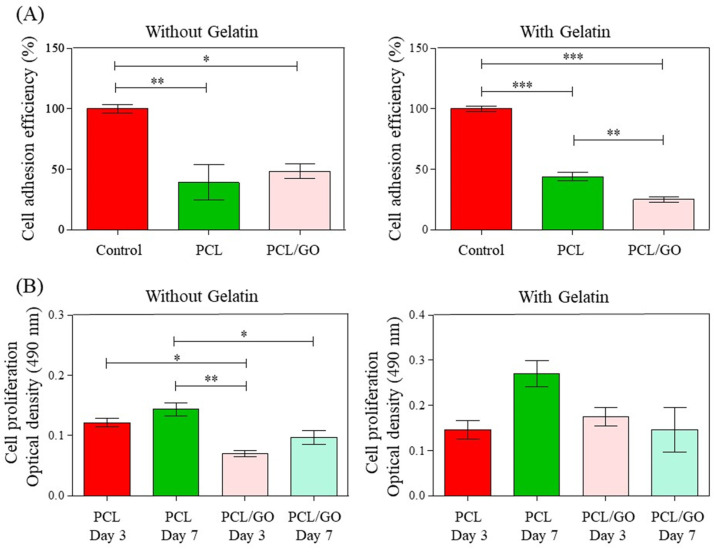
In vitro biological characterization of electrospun PCL and PCL/2 wt% GO without and with gelatin using hGMSCs. (**A**) Cell adhesion efficiency and (**B**) cell proliferation efficiency (n = 3, * *p* ≤ 0.05, ** *p* ≤ 0.01, and *** *p* ≤ 0.001).

**Table 1 polymers-16-00129-t001:** Data from the TGA analysis. T10, Tmax, and gelatin content in the PCL/Gt, PCL/1 wt% GO/Gt, and PCL/2 wt% GO/Gt scaffolds.

Sample	T10 * (°C)	Tmax ** (°C)	Gelatin Content (%)
Gt	244	318	--
PCL	363	388	--
PCL/1 % GO	367	388	--
PCL/2 % GO	375	404	--
PCL/Gt	362	389	0.9
PCL/1 % GO/Gt	346	387	14.3
PCL/2 % GO/Gt	312	407	21.6

* T10: Temperature at 10% of weight loss. ** Tmax: Temperature at maximum rate of weight loss.

## Data Availability

The data presented in this study are available on request from the corresponding author.

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
