# Peer review of "Effect of Gelatin Coating and GO Incorporation on the Properties and Degradability of Electrospun PCL Scaffolds for Bone Tissue Regeneration"

_polymers, 2023, doi:10.3390/polym16010129_

Round 1

Reviewer 1 Report

Comments and Suggestions for Authors

This manuscript proposed gelatin-coated electrospun PCL/GO scaffolds for bone tissue engineering. The authors have done lots of work on characterization, but the novelty and feasibility of this study are big concerns. As the authors have summarized in the introduction, many studies have been done using PCL/GO/gelatin, so I didn't see this work's necessity. In addition, in this work gelatin is not anchored on the scaffold using any chemical method, and gelatin itself could dissolve in the warm medium. It could be assumed that once the gelatin-coated scaffold is placed in the incubator or in vivo, the gelatin on the surface could dissolve away, which is a flaw of the biomaterial design. 

Some statement in the manuscript is not correct, for example, in line 42, cells cannot attached on hydrophobic materials. 

The introduction part is tedious and not well-organized.

Author Response

Review 1 We appreciate your suggestions, these are of great importance to improve the manuscript. Each of the suggestions was made in as much detail as possible. In the manuscript the suggestions are highlighted in yellow color. Comment 1: This manuscript proposed gelatin-coated electrospun PCL/GO scaffolds for bone tissue engineering. The authors have done lots of work on characterization, but the novelty and feasibility of this study are big concerns. As the authors have summarized in the introduction, many studies have been done using PCL/GO/gelatin, so I didn't see this work's necessity. In addition, in this work gelatin is not anchored on the scaffold using any chemical method, and gelatin itself could dissolve in the warm medium. It could be assumed that once the gelatin-coated scaffold is placed in the incubator or in vivo, the gelatin on the surface could dissolve away, which is a flaw of the biomaterial design. Answer: Thanks to Review 1 for the feedback as in our original manuscript the novelty was not fully explained. We changed the introduction focusing on the new approach of our design. In particular, previous results studying PCL/GO/Gt ternary samples were focused on mixing the components (Gt into the bulk of the PCL/GO composite). Our results otherwise show for a first time a Gt coating on a PCL/GO scaffold focusing on the effect of GO incorporation on the amount of Gt absorbed. The latter is relevant as it is known that GO into polymer matrices change the hydrophobicity increasing the absorption of biomolecules and biopolymers. For this reason, a surface modification of the surface of PCL and a crosslinking process were not carried out. In the last part of the introduction section these points were added. As the amount of Gt was increased, the effect of Gt on the PCL/GO surface was extended over time, and for that reason the effect of Gt still was observed after 60 days according to water absorption results (Figure 8b). Furthermore, an examination of samples without gelatin coating revealed a noticeable reduction in both water absorption and degradability. This observation underscores the significant surface modification of PCL due to intermolecular forces between PCL and Gt, which endure over time. On the other hand, cell proliferation test showed that Gt had an effect after 7 days of tests on the scaffolds. It is clear that our approach did not avoid the dissolution of Gt, as the Reviewer 1 stated, and for that reason is part of our discussion. Comment 2: Some statement in the manuscript is not correct, for example, in line 42, cells cannot attached on hydrophobic materials. Answer: The Reviewer 1 was right as the text was not clear. In the original text the concept “optimal hydrophobicity” was associated with “low hydrophobicity”. The text was modified. Comment 3: The introduction part is tedious and not well-organized. Answer: The introduction was shortened allowing to express our main ideas directly avoiding tedious parts. The new part associated with “gelatine coating” and the novelty of our approach should improve the organization of this section.

Reviewer 2 Report

Comments and Suggestions for Authors

Comments for polymers-2778375

1.      It is suggested to use SEM to capture the morphology and size of the original GO powder.

2.      It is suggested to use SEM to capture the distribution morphology of GO within PCL fibers.

3.      Several papers exist on binary or ternary composite electrospun fiber scaffolds of PCL, gelatin, and GO. Discuss these references in the introduction, clearly outlining the distinctions between your paper and these works, emphasizing the innovations in your paper.

4.      It is suggested to cite more literature on graphene oxide improving the mechanical properties and cell responses of polymers, for instance, https://doi.org/10.1016/j.jmbbm.2023.105848

5.      Please clarify the basis for selecting GO content at 1% and 2%—was it based on preliminary experiments or referenced literature?

6.      Please kindly explain the rationale behind choosing a GT concentration of 2.5% w/v—was it based on preliminary experiments or referenced literature?

7.      Some images have low clarity/resolution, and the font size of labeling text is small. Please review each one individually to enhance quality.

8.      In Figure 6, was the water absorption rate of PCL tested? What were the average values and standard deviation?

9.      In Figure 11, cell adhesion and proliferation of PCL/GO are higher than PCL, while cell adhesion of PCL/GO/GT is lower than PCL/GT. Please provide further explanation and discussion on these results.

10.  Please improve English language grammar.

Comments on the Quality of English Language

Moderate editing of English language required.

Author Response

Dear Review 2

We appreciate your suggestions, these are of great importance to improve the manuscript. Each of the suggestions was made in as much detail as possible. In the manuscript the suggestions are highlighted in yellow color.

Comments for polymers-2778375

  1. It is suggested to use SEM to capture the morphology and size of the original GO powder.

Answer: The section 3.1 has been added in order to show the characterization of GO. Whitin this section, an exposition of SEM, TEM and XDR analyses have been provided incorporated into the lines 301-314

GO was morphologically characterized by SEM and TEM. Figure 1A shows the SEM image, which revealed that GO exhibited a distinctly rougher surface and had thin sheets arranged in a random and overlapped fashion. Figure 1B shows the TEM image, which provided additional insights by showcasing a single, transparent, and wrinkled sheet of GO. The morphology of our product was quite consistent with that previously reported [43-46].

The structural information of GO shown in Figure 1C was obtained from the XRD spectra. The XDR GO diffractogram showed two characteristic peaks at 2θ=11.0° and 42.3°, corresponding to the reflections from the (002) and (101) planes [47,48]. The interlayer spacing was calculated at 0.80 nm.

  1. It is suggested to use SEM to capture the distribution morphology of GO within PCL fibers.

Answer: Efforts were made to elucidate the distribution morphology of GO within PCL fibers through SEM imaging; however, distinguishing GO within PCL proved challenging. Consequently, we conducted a comprehensive Raman spectroscopy analysis, as detailed in Section 3.2 (lines 351-376), to substantiate the incorporation of GO into the scaffolds:

Raman shifts of a) PCL, b) PCL/2 wt% GO obtained after focusing the confocal on the sample surface are presented in Figure 4. Several peaks characteristic of PCL are observed, the more prominent at 916 cm−1(νC–COO), and others within the spectral ranges 1003–1110 cm−1 (skeletal stretching), 1270–1300 cm−1 (ωCH2), 1405–1470 cm−1(δCH2) and 2800–3200 cm−1 (νCH) are referred to the crystalline fraction [54]. However, the characteristic Raman signal of GO is not detectable in the Raman spectrum with this methodology (See Figure 4b).

To reveal GO-Raman signal an enhancement strategy was employed, shell isolated nanoparticles (SHINs) allow to enhance the Raman signal of GO by coupling with GO groups (preparation and details will be given in Enhanced Raman spectroscopy applied to the study of water-graphene interface, to be published by A. Maine, L. Caballero and F. Melo). SHINS where deposited onto the sample surface and let them to dry.  The confocal microscope was then focused on the surface where GO spectrum was maximum, Figure 4c.  Thus, GO Raman spectrum was observed, which allowed to confirm the presence of GO in the samples. The main lines of GO corresponding to D and G peaks are clearly visible [55]. Furthermore, in the Figure S2, analogous outcomes are observe in the spectra of PCL/1wt% GO, corroborating the incorporation of GO.

In addition, Figure 4d, is slightly out of focus which permits the observation of both Raman spectra, that of GO and PCL.

  1. Several papers exist on binary or ternary composite electrospun fiber scaffolds of PCL, gelatin, and GO. Discuss these references in the introduction, clearly outlining the distinctions between your paper and these works, emphasizing the innovations in your paper.

Answer: Thanks to Review 2 for the feedback as in our original manuscript the novelty was not fully explained. We changed the introduction focusing on the new approach of our design. In particular, previous results studying PCL/GO/Gel ternary samples were focused on mixing the components (Gel into the bulk of the PCL/GO composite).

The introduction was shortened allowing to express our main ideas directly avoiding tedious parts. The new part associated with “gelatine coating” and the novelty of our approach should improve the organization of this section.

The next paragraph was incorporated into the lines 96-112:

Examining the performance of higher concentrations (1 wt.%) of GO is crucial for developing scaffolds with enhanced thermal and mechanical properties, all while ensuring non-toxicity. In addition to seeing the effect on degradability over time.

Beside mixing PCL with biopolymers or nanoparticles, another strategy to improve its bioactivity toward BTE is by a coating, in particular with Gt, as it provides desirable biological properties to the bulk polymer retaining its structural properties [27,28]. A electrospinned PCL scaffold coated with Gt presents improved mechanical behavior as compared with PCL/Gt composites, and higher bioactivity than pure PCL, although lower than composites [27]. Three routes are explored to coat Gt on PCL: direct coating, coating on a surface modified PCL (for instance by plasma treatment) and coating followed by a crosslinking Gt reaction [28-30]. Crosslinking and surface modification processes are currently explored to delay the high solubility of Gt in aqueous solution, being the former the most used although it can increase the toxicity of the scaffolds [31], and it does not avoid completely the solubility of Gt [32]. Although the different functional groups of GO can increase the polarity of PCL, and therefore the adsorption of biomolecules and biopolymers, there is not studies about the effect of this filler on the Gt coating characteristics of PCL/GO scaffolds.

  1. It is suggested to cite more literature on graphene oxide improving the mechanical properties and cell responses of polymers, for instance, https://doi.org/10.1016/j.jmbbm.2023.105848

Answer: The discussion of mechanical properties was improved. The new sentences were included and highlighted in yellow color (Lines 555-558).

Guo et al. reported an increase in Young’s Modulus of ca. 35.6% for PLA/GO (0.1 wt.%) compared to the neat PLA obtained by 3D printing. The authors mentioned that the mechanical properties of polymer nanocomposite are related to the content and distribution of the filler [57].

The next sentences were included in Lines 563-587

Considering this, the stiffer behavior of the PCL/GO electrospun mats could also be related to a decrease in fiber diameter.

For Gt-coated scaffolds, the Young’s modulus did not change in comparison with neat PCL, but the value decreased in comparison with PCL/GO. The reinforcement effect of GO was not appreciable and these reductions were mainly attributed to a decrease in the crystallinity due to the gelatin coating (Figure 10B) [76]. The gelatin coating process was executed at 37°C, and the onset of the melting process occurs slightly before this temperature. Although the samples did not undergo complete melting at this threshold, they experienced a reduction in crystallinity. This phenomenon is likely due to the higher mobility of polymeric chains, probably enabling the accumulation of GO nanolayers within the fibers during treatment. The resultant decline in crystallinity, coupled with the accumulation of GO, contributes to a reduction in the mechanical properties, encompassing Young's modulus, strain at break, and tensile strength. This effect mirrors a similar annealing phenomenon observed in prior studies [76].

Different results have been reported for electrospun PCL/GO/Gt. The incorporation of GO (1, 1.5, and 2 wt%) increased tensile strength and Young's modulus by ca. 117% and 128%, respectively, when 1.5% graphene was incorporated into PCL/Gt nanofibers, compared to the values of PCL/Gt nanofibers. The authors indicated that gelatin could act as a compound for graphene functionalization or bridge the attachment of PCL and graphene [26]. The tensile strength did not change with the GO incorporations, as expected for PCL composites with these amounts of GO, as previously reported [21]. However, PCL/Gt and PCL/GO/Gt showed decreased tensile strength in comparison with the neat PCL. Unal et al. prepared PCL and PCL-Gt and showed that Gt incorporation reduced the mechanical strength of the composite, an effect attributed to the weak physical properties of gelatin [21].

  1. Please clarify the basis for selecting GO content at 1% and 2%—was it based on preliminary experiments or referenced literature?

Answer:

The selection of GO concentrations at 1% and 2% was based on a comprehensive literature review and preliminary experimentation. Previous studies by Heidari M. et al (2017) and Francavilla P. et al (2021) indicated that the maximum GO concentration used in similar contexts was 2%. Moreover, our preliminary experiments revealed challenges associated with higher concentrations. At GO concentrations above 2%, the PCL solution exhibited high viscosity, leading to difficulties in obtaining fibers, resulting in the formation of droplets in some instances. Furthermore, in other cases, the electrospinning process was hindered due to the solution's resistance to being expelled through the metal needle.

To statistically investigate the impact of GO concentration on scaffold properties, we chose to include a central point at 1% GO concentration. Our experiments, using 1% and 2% as concentration levels, demonstrated a clear correlation between increasing GO concentration and various biomaterial properties. Notably, as the GO concentration increased, we observed a reduction in fiber size and an increased in the Mechanical properties, accompanied by an increase in water absorption and degradability. Consequently, we opted not to test with other intermediate concentrations.

References:

Heidari, M., Bahrami, H., & Ranjbar-Mohammadi, M. (2017). Fabrication, optimization and characterization of electrospun poly (caprolactone)/gelatin/graphene nanofibrous mats. Materials Science and Engineering: C, 78, 218-229.

Francavilla, P., Ferreira, D. P., Araújo, J. C., & Fangueiro, R. (2021). Smart fibrous structures produced by electrospinning using the combined effect of pcl/graphene nanoplatelets. Applied Sciences, 11(3), 1124.

  1. Please kindly explain the rationale behind choosing a GT concentration of 2.5% w/v—was it based on preliminary experiments or referenced literature?

Answer:

After conducting preliminary experiments, we determined that a gelatin concentration of 2.5% w/v would be the most suitable for our study. Our observations revealed that gelatin absorption remained consistent, irrespective of variations in the initial gelatin concentration within the solution. The extent of gelatin absorption is found to be intricately linked to fiber morphology and primarily influenced by the GO content, as highlighted in Table 1. To maintain simplicity and focus solely on the influence of GO on biomaterial properties, we opted for a single gelatin solution at a concentration of 2.5%. This decision was made to distinctly showcase the effects of GO in the biomaterials obtained.

  1. Some images have low clarity/resolution, and the font size of labeling text is small. Please review each one individually to enhance quality.

Answer:

You are right, the clarity and resolution of images were improved (particularly Figures 2, 5, 8-10)

  1. In Figure 6, was the water absorption rate of PCL tested? What were the average values and standard deviation?

Answer:

The water absorption of PCL was 15 ± 11%, this is already on the graph, but it is too low compared to the other materials due to its semicrystalline structure and hydrophobic behavior.

  1. In Figure 11, cell adhesion and proliferation of PCL/GO are higher than PCL, while cell adhesion of PCL/GO/GT is lower than PCL/GT. Please provide further explanation and discussion on these results.

Answer:

Thank for your comments, as per reviewer comment we provide a further explanation about cell adhesion we have changed and updated the original text to the following sentences in lines 677 to 705.

The gelatin coating did not affect the initial adhesion of hGMSCs to the PCL scaffolds. There were no significant differences in cell adhesion to the PCL/GO/Gt scaffolds compared to PCL in the samples without gelatin. However, when gelatin coating was incorporated, cell adhesion to PCL/GO decreased.

In the adhesion process, the presence of gelatin with GO significantly decreased the colorimetric signal (Figure 13 A, right). As previously reported, there are differences in the proliferation of MSCs depending on the concentration of gelatin used in the culture. For instance, in a similar study, Khan et al. used adipose-derived mesenchymal stem cells (Ad-MSCs), cultured them in vitro with different concentrations of gelatin during 24 hrs [83], and found that 0.5% gelatin significantly increased the proliferation rate, as evaluated by a colorimetric assay. Conversely, 2% and 4% gelatin decreased cell proliferation significantly. In the current study we used 2.5% gelatin and observed the same trend at 24 hrs; namely, a decreased colorimetric signal, which indicates a decrease of both adhesion and proliferation. Furthermore, the current research is a preliminary study to evaluate the biocompatibility effect of Gt on the PCL/GO scaffolds and the current results will help us to establish further experimental designs.

In addition, high concentrations of GO can affect cell viability [73]. Since gelatin is viscous and has low solubility [84], we speculate that reactive oxygen species (ROS) could be trapped within the gelatin matrix, negatively affecting cell viability in PCL/GO/Gt scaffolds. Future live/dead staining experiments will be considered to test this hypothesis and optimize GO and gelatin concentrations to evaluate their biological effects.

In turn, the effects of Gt and GO disappear during the first 24 hours (adhesion). Therefore, cell adhesion of MSCs to a scaffold with GO and Gt would involve fixation, extension or propagation mechanisms, organization of the actin cytoskeleton, and formation of focal adhesions. To the best of our knowledge, there are no studies that have specifically analyzed all these processes according to molecular interactions, and this interesting approach could be considered for future experiments to precisely determine the interaction mechanisms of MSCs under these conditions.

10.Please improve English language grammar. Comments on the Quality of English Language Moderate editing of English language required.

Answer:  The English was reviewed by a native English speaker

Reviewer 3 Report

Comments and Suggestions for Authors

This manuscript described the study of incorporating graphene oxide and gelatin coating to improve the properties of electrospun PCL sheet. Physical, mechanical and biological characterizations were carried to evaluate the effect of such modifications. The experimental was carefully designed and carried out, but some revisions should be made to improve the quality of the manuscript.

- The title should be considered revising to be the effect of both GO and gelatin coating since the results were compared in terms of both GO addition, Gelatin coating and GO+gelatin coating with PCL.

- Section 2.6: Recheck the accuracy of the statement "The analysis focused on the effects of two parameters: electrospinning voltage (15 kV and 20 kV) and GO concentration (0%, 1%, and 2%), for evaluating fiber diameter and pore size.". I did not see such voltage change and the gelatin coating effect was missing.

-P. 9, Fig 4: Authors mentioned about the confirmation of the presence of Gt, but did not report anything about GO confirmation or change in IR. Addition is needed.

-P10, table 1: Method used to determine the content of gelatin in PCl and PCL/GO should be clearly explained. It was rather vague at presence without clear details.

-P 12: Authors mentioned about two PCL degradation mechanisms, but failed to explained the mechanism of degradation mechanism of electrospun PCL or PCL/GO, PCL/GO/Gt in this study whether following which mechanism or others.

- Section 3.7: Authors briefly reported and discussed about the effect of gelatin coating o tensile properties. Nothing was clearly discussed about the reasons on why decreases were seen for gelatin coated samples especially for PCL/GO samples compared to uncoated ones. Also the changes differed among modulus, strength and strain. What were the reasons behind all of these. Authors briefly discussed about the decrease in crystallinity in gelatin coated samples, but there was also the effect of GO which clearly increased the tensile properties and was not expected to be countered by such crystallinity decrease.

-P 14, authors mentioned that Lui et al. demonstrated that graphene oxide and PCL alone were not capable of inducing hydroxyapatite mineralization [20]. However, authors failed to explained further whether this was similar of different from this study.

-P. 15: authors mentioned that the spectra obtained by energy dispersive X-ray spectroscopy (EDX) coupled to SEM (Figure 10) showed that the spherical structures present on the scaffold fibers effectively corresponded to hydroxyapatite. I do not believed that EDX and SEM could showed that such particles were hydroxyapatite. To be certain, XRD should be done to confirm instead of ATR-FTIR which was not specific to phase determination.

-P 16, fig 11: authors did not clearly explained the effect of gelatin coating on cell proliferation. Why cells could proliferate from 3 to 7 days for PCL/GO, but decreased for gelatin coated PCL/GO/Gt from 3 to 7 days.

-Discussion is needed to be improved since it was like simply quoting references without further delineating the results.

-Conclusions: It was rather a brief results, not a conclusion. Revision is needed.

Comments on the Quality of English Language

Please recheck the spelling errors which could still be ound.

Author Response

Dear Review

We appreciate your suggestions, these are of great importance to improve the manuscript. Each of the suggestions was made in as much detail as possible. In the manuscript the suggestions are highlighted in Yellow color.

This manuscript described the study of incorporating graphene oxide and gelatin coating to improve the properties of electrospun PCL sheet. Physical, mechanical and biological characterizations were carried to evaluate the effect of such modifications. The experimental was carefully designed and carried out, but some revisions should be made to improve the quality of the manuscript.

  1. The title should be considered revising to be the effect of both GO and gelatin coating since the results were compared in terms of both GO addition, Gelatin coating and GO+gelatin coating with PCL.

Answer: Thanks for your comments, to improve visibly the aims and experimental parameters  of our research work, the new title of article will be:

Effect of gelatin coating and GO incorporation on the properties and degradability of electrospun PCL scaffolds for bone tissue regeneration

  1. Section 2.6: Recheck the accuracy of the statement "The analysis focused on the effects of two parameters: electrospinning voltage (15 kV and 20 kV) and GO concentration (0%, 1%, and 2%), for evaluating fiber diameter and pore size.". I did not see such voltage change and the gelatin coating effect was missing.

Answer: You have reason, the sentence was change and improvement in the lines 296-298:

 The analysis focused on the effects of two parameters: GO concentration (0%, 1%, and 2%) ,and gelatin coating for evaluating degradation, mechanical, thermal and biological properties.

  1. 9, Fig 4: Authors mentioned about the confirmation of the presence of Gt, but did not report anything about GO confirmation or change in IR. Addition is needed.

Answer:

Thanks for your recommendation, effectively in the results section, it was can not possible to discussed deeply the GO incorporation due to that the characteristic signals were very difficult to assign in the spectra reported in Figure 6-A.  The low quantity of GO resulted in signal overlap with PCL signals, making the distinction difficult. Nonetheless, we were able to supplement the analysis by conducting an Raman spectroscopy analysis, as outlined in Section 3.2 (lines 351-376), to provide evidence of the successful incorporation of GO into the scaffolds."

Raman shifts of a) PCL, b) PCL/2 wt% GO obtained after focusing the confocal on the sample surface are presented in Figure 4. Several peaks characteristic of PCL are observed, the more prominent at 916 cm−1(νC–COO), and others within the spectral ranges 1003–1110 cm−1 (skeletal stretching), 1270–1300 cm−1 (ωCH2), 1405–1470 cm−1(δCH2) and 2800–3200 cm−1 (νCH) are referred to the crystalline fraction [54]. However, the characteristic Raman signal of GO is not detectable in the Raman spectrum with this methodology (See Figure 4b).

To reveal GO-Raman signal an enhancement strategy was employed, shell isolated nanoparticles (SHINs) allow to enhance the Raman signal of GO by coupling with GO groups (preparation and details will be given in Enhanced Raman spectroscopy applied to the study of water-graphene interface, to be published by A. Maine, L. Caballero and F. Melo). SHINS where deposited onto the sample surface and let them to dry.  The confocal microscope was then focused on the surface where GO spectrum was maximum, Figure 4c.  Thus, GO Raman spectrum was observed, which allowed to confirm the presence of GO in the samples. Main lines of GO corresponding to D and G peaks are clearly visible [55]. Furthermore, in the Figure S2, analogous outcomes are observe in the spectra of PCL/1wt% GO, corroborating the incorporation of GO.

In addition, Figure 4d, is slightly out of focus which permits the observation of both Raman spectra, that of GO and PCL.

  1. P10, table 1: Method used to determine the content of gelatin in PCl and PCL/GO should be clearly explained. It was rather vague at presence without clear details.

Answer: The methodology for calculating gelatin content was introduced in Section 2.5.4 (lines 218-224) as follows:

The Gt content of the scafolds was determined by comparing the residual weights at 600 °C of each material and using the following equations [35]

(Equation 4)

(Equation 5)

Where Rt (%)i0 represents the residual weight percentage of the i-th component, WfiS is the weight fraction of the i-th component in the scafold, and Rt (%)total is the residual weight percentage of the electrospun mat with all components.

  1. P 12: Authors mentioned about two PCL degradation mechanisms, but failed to explained the mechanism of degradation mechanism of electrospun PCL or PCL/GO, PCL/GO/Gt in this study whether following which mechanism or others.

Answer: Thanks for your recommendation, to leave to more clearly as possible the hydrolytic degradation mechanism of the PCL, PCL/GO and PCL/GO/Gelatin nanocomposites, and for other side, the possible effects of GO and GT in this mechanism, the next sentence was incorporated into the manuscript in lines 510-518.

In summary, in the PCL and PCL/GO systems, the degradation was very low, as shown in Figure 9A. For these systems, the degradation process probably occurs on the surface, and due to the high hydrophobicity of the PCL matrix, the degradation is clearly delayed. On the other hand, there was an increase in the size of the surface pores of the fibers in PCL/Gt and PCL/GO/Gt (Figure 9B), which was greater for the PCL/GO/Gt system. This increase indicates the development of a surface erosion process associated with the presence and release of gelatin, which has greater hydrophilicity and allows greater interaction between the fibers and water molecules, thus accelerating the intramolecular hydrolysis processes of the PCL matrix and breaking the fibers.

6.Section 3.7: Authors briefly reported and discussed about the effect of gelatin coating o tensile properties. Nothing was clearly discussed about the reasons on why decreases were seen for gelatin coated samples especially for PCL/GO samples compared to uncoated ones. Also the changes differed among modulus, strength and strain. What were the reasons behind all of these. Authors briefly discussed about the decrease in crystallinity in gelatin coated samples, but there was also the effect of GO which clearly increased the tensile properties and was not expected to be countered by such crystallinity decrease.

Answer: The discussion regarding the variation of mechanical properties in relation to gelatin coating was inadvertently omitted. Consequently, the following paragraph has been incorporated into the manuscript (lines 566-576):

For Gt-coated scaffolds, the Young’s modulus did not change in comparison with neat PCL, but the value decreased in comparison with PCL/GO. The reinforcement effect of GO was not appreciable and these reductions were mainly attributed to a decrease in the crystallinity due to the gelatin coating (Figure 10B) [76]. The gelatin coating process was executed at 37°C, and the onset of the melting process occurs slightly before this temperature. Although the samples did not undergo complete melting at this threshold, they experienced a reduction in crystallinity. This phenomenon is likely due to the higher mobility of polymeric chains, probably enabling the accumulation of GO nanolayers within the fibers during treatment. The resultant decline in crystallinity, coupled with the accumulation of GO, contributes to a reduction in the mechanical properties, encompassing Young's modulus, strain at break, and tensile strength. This effect mirrors a similar annealing phenomenon observed in prior studies [76].

  1. P 14, authors mentioned that Lui et al. demonstrated that graphene oxide and PCL alone were not capable of inducing hydroxyapatite mineralization [20]. However, authors failed to explained further whether this was similar of different from this study.

Answer:  The next sentence was incorporated into the lines 608-611

Lui et al. demonstrated that graphene oxide alone was not capable of inducing hydroxyapatite mineralization after immersion of GO in SBF after 14 days [15]. The results, verified by SEM, FTIR, and EDX analysis showed only a trace amount of calcium.

  1. P. 15: authors mentioned that the spectra obtained by energy dispersive X-ray spectroscopy (EDX) coupled to SEM (Figure 10) showed that the spherical structures present on the scaffold fibers effectively corresponded to hydroxyapatite. I do not believed that EDX and SEM could showed that such particles were hydroxyapatite. To be certain, XRD should be done to confirm instead of ATR-FTIR which was not specific to phase determination.

Answer: Thanks for your suggestion, you are correct. To enhance conciseness in verifying hydroxyapatite (HA) formation, we have included X-ray diffraction (XRD) analysis in the discussion section, specifically addressed in lines 642 to 646. However, the XRD patter spectra figure will be included in supplementary file section (Figure S3):

Finally, alongside SEM, EDX, and FT-IR analysis, the growth of HA crystals was corroborated by X-ray diffraction (see Supplementary Figure S3). Notably, for PCL/2%GO/Gt nanocomposites, after 28 days of immersion in SBF solution, a distinct peak at 32 ° was observed, corresponding to the (211) plane of the hexagonal HA nanocrystalline phase [79].

  1. P 16, fig 11: authors did not clearly explained the effect of gelatin coating on cell proliferation. Why cells could proliferate from 3 to 7 days for PCL/GO, but decreased for gelatin coated PCL/GO/Gt from 3 to 7 days.

Answer: Thanks for the feedback. As per reviewer comment, we agree that the average signal tends to decrease in gelatin-coated PCL/GO/Gt from 3 to 7 days. However, this difference is not statistically significant, and we observe that in both cases, the dispersion of the data with Gt in this experimental condition is greater.  Furthermore, as we mentioned earlier in the study, high concentrations of GO can affect cell viability. Since gelatin is viscous and has low solubility [84], we speculate that reactive oxygen species (ROS) could be trapped within the gelatin matrix, negatively affecting cell viability in PCL/GO/Gt scaffolds. Future live/dead staining experiments will be considered to test this hypothesis and so GO and gelatin concentrations could be optimized to evaluate biological effects.

The following text was updated in the manuscript in the lines 694 to 698

In addition, high concentrations of GO can affect cell viability [73]. Since gelatin is viscous and has low solubility [84], we speculate that reactive oxygen species (ROS) could be trapped within the gelatin matrix, negatively affecting cell viability in PCL/GO/Gt scaffolds. Future live/dead staining experiments will be considered to test this hypothesis and optimize GO and gelatin concentrations to evaluate their biological effects.

  1. Discussion is needed to be improved since it was like simply quoting references without further delineating the results.

Answer: A comprehensive review of the entire discussion section has been conducted. We ensured that the revised discussion offers a clear interpretation of the results, highlights key observations, and establishes meaningful connections with relevant literature to enhance the overall quality of the manuscript.

  1. Conclusions: It was rather a brief results, not a conclusion. Revision is needed.

Answer: The conclusions was improved and incorporated in the new manuscript:

Electrospun fiber scaffolds based on PCL/GO were prepared and coated with Gt for bone regeneration. Our results showed that the presence of GO not only decreased fiber diameter and increased scaffold stiffness but also improved the gelatin coating process. The Gt adsorption increased from 0.9 wt% for pure PCL to 21.6 wt% for the PCL/GO with 2 wt%, without the need of any surface treatment of the fibers. Our ap-proach to improve the Gt coating by mixing PCL with GO was an interesting route to promote the bioactivity capacity of PCL, as the coated composites showed hydroxyap-atite crystals after soaking in SBF solution. In addition, the in vitro biological analysis using hGSMCs demonstrated that uncoated and coated PCL andPCL/GO samples were viable substrates for cell adhesion and that Gt coating improved cell proliferation. These results introduce a new route to produce Gt-coated scaffolds taking advantage of the properties of GO as a filler.

Round 2

Reviewer 1 Report

Comments and Suggestions for Authors

All comments have been addressed.

Reviewer 3 Report

Comments and Suggestions for Authors

Authors have revised the manuscript accordingly.

Comments on the Quality of English Language

NA